# Sociodemographic factors, biomarkers and comorbidities associated with post-acute COVID-19 sequelae in UK Biobank

Marta Alcalde-Herraiz[1], Shahed Iqbal[2], Jeffrey J. Wallin[2], Yunhao Liu[2], Wildaliz Nieves[2], Mark Berry[2], Marti Catala[1], Daniel Prieto-Alhambra [1,3] & Junqing Xie [1]

Long-term sequelae of COVID-19 remain critical public health concerns, with limited therapeutic options available. We conducted two case-control studies among COVID-19 infected individuals in the UK Biobank to explore the association of sociodemographic factors, clinical biomarkers, and comorbidities with the risk of two key phenotypes: Long COVID (LC, defined by patient self-report symptoms) and post-acute complications of SARS-CoV-2 infection (PACS, defined by clinical diagnosis), separately. Our study included 8,668 participants in the LC cohort (32% classified as cases) and 108,407 in the PACS cohort (with 2% being cases). Findings showed that age and sex were associated with both LC and PACS but in opposite directions. Additionally, obesity, socioeconomic deprivation, elevated C-reactive protein, triglyceride, vitamin D, HbA1c, cystatin C, urate, and alanine aminotransferase, and decreased HDL cholesterol and IGF-1, as well as CKD and COPD, were associated with LC. Most of these factors were also significant for PACS, except for alanine aminotransferase and vitamin D. These findings have potential mechanistic implications for the distinction between LC and PACS and can guide clinical implementation of identifying high-risk groups for targeted vaccination or other public health mitigation strategies.

Even though COVID-19 cases and deaths have significantly decreased globally, long-term health consequences of SARS-CoV-2 infection, commonly known as Post-COVID-19 Conditions (PCC), are emerging as a critical public health issue[1,2]. An estimated 3% of the UK population experienced symptoms for at least four weeks after infection[3], with risk for PCC potentially not solely dependent on the severity of the acute COVID-19 infection.

PCC can be classified into two main categories[4–6]: Long COVID (LC) and post-acute complications of SARS-CoV-2 infection (PACS). LC is defined by the persistence or onset of COVID-19-related symptoms beyond one to three months after the initial infection. Common symptoms include fatigue, shortness of breath, and other symptoms that can significantly impact day-to-day functioning. In contrast, PACS typically refers to more severe complications emerging in the same time frame, such as thromboembolic or cardiovascular events, including angina, myocardial infarction, or pulmonary embolism.

Previous studies have aimed to characterise patients with LC[7]. Tsampasian et al. conducted a systematic review and meta-analysis of 41 studies to explore the risk factors of PCC in adult patients. Sociodemographics such as female sex, age, high BMI, and smoking were associated with an increased risk of PCC. The presence of comorbidities like anxiety or depression, asthma, chronic obstructive pulmonary disease (COPD), diabetes, immunosuppression, and ischaemic heart disease was also linked to an increased risk for PCC.

[1]Centre for Statistics in Medicine, NDORMS, University of Oxford, Oxford, UK. [2]Gilead Sciences, Inc., Foster City, USACA. [3]Department of Medical Informatics, Erasmus University Medical Center, Rotterdam, The Netherlands. ✉e-mail: daniel.prietoalhambra@ndorms.ox.ac.uk

Identifying potential biomarkers for new conditions is key for an early detection, better understanding of pathophysiology, and the development of effective treatments. Several biomarkers have been identified to date that can contribute to the study and management of LC. For example, people with prolonged symptoms have shown higher levels of inflammatory markers, like C-reactive protein and interleukin-6, which points to ongoing low-grade inflammation[8]. Separately, biomarkers tied to endothelial dysfunction, such as vascular adhesion molecules and von Willebrand factor, have been reported in association with LC[9]. However, the heterogeneous nature of LC and PACS needs more intensive investigation, ideally in larger populations and granular datasets. We therefore leveraged linked UK Biobank data to perform a hypothesis-free analysis and explore the associations and interplay between sociodemographic factors, biomarkers, and comorbidities and the risk of LC and PACS.

## Results

### Study cohorts
See Fig. 1 for more details on the number of individuals for each linked dataset.

Out of the 275,234 UK Biobank participants that had a valid linkage with COVID-19 surveillance data, 46,793 had a positive test result and had answered all the questions from the Health and Well-Being online questionnaire. Among these participants, 8668 fulfilled the selection criteria to be part of the LC base cohort. A total of 2751 (32%) reported at least one symptom when answering the questionnaire and hence were classified as LC cases. The remaining 5917 (68%) were classified as LC controls (Supplementary Fig. 1A).

In total, 115,007 UK Biobank participants had a valid linkage to Hospital Episode Statistics (HES) data and to the COVID-19 surveillance data, and a positive COVID-19 test. Of these, 108,407 fulfilled the selection criteria to be part of the PACS base cohort. A total of 1940 (2%) had at least one PACS diagnosis beyond 30 days after infection and were therefore classified as PACS cases, whereas 106,467 (98%) were classified as PACS controls (Supplementary Fig. 1B).

Baseline characteristics of both study cohorts stratified by case-control status are reported in Tables 1–3. LC cases had a higher proportion of women compared to LC controls (54% and 52%, respectively) and appeared to live more often in socio-economically deprived areas (index of multiple deprivation (IMD) of 15 and 13, respectively). PACS cases were older compared to PACS controls, with a mean age of 72 and 67, respectively, and with a higher proportion of men (57% and 44%, respectively).

### Linearity assessment
Results of the exploratory data analysis, data curation and correlation analysis are reported in Supplementary Note 1.

We explored potential non-linear relationships between each biomarker and the study outcomes using natural cubic spline curves. For C-reactive protein and HDL cholesterol, the non-linear cubic spline model provided a better fit for the logit function of LC compared to a linear model (see Fig. 2A). For the other biomarkers, the linear logistic regression model was a better fit. Similarly, we identified a potential non-linear relationship for 12 out of the 17 biomarkers with PACS (Fig. 2B).

### Variable selection
LASSO regularisation identified all the sociodemographic factors, all the biomarkers and 9 out of 17 comorbidities as associated with LC (Supplementary Table 1). For PACS, LASSO regularisation selected all sociodemographic factors, 15 out of 17 biomarkers, and 15 out of 17 comorbidities (see Supplementary Table 2).

### Outcome model regression analyses
For LC, none of the variables exhibited high levels of multicollinearity (see Supplementary Table 3). Full results, including crude and adjusted odds ratios (ORs) are detailed in Fig. 3.

Regarding socio-demographics, younger individuals (<55) were associated with an increased risk of LC compared to older people (≥75) in the adjusted analysis (OR$_{adjusted}$ = 1.23, 95% CI = 1 to 1.42). Obesity and deprivation were linked to an increased risk of LC

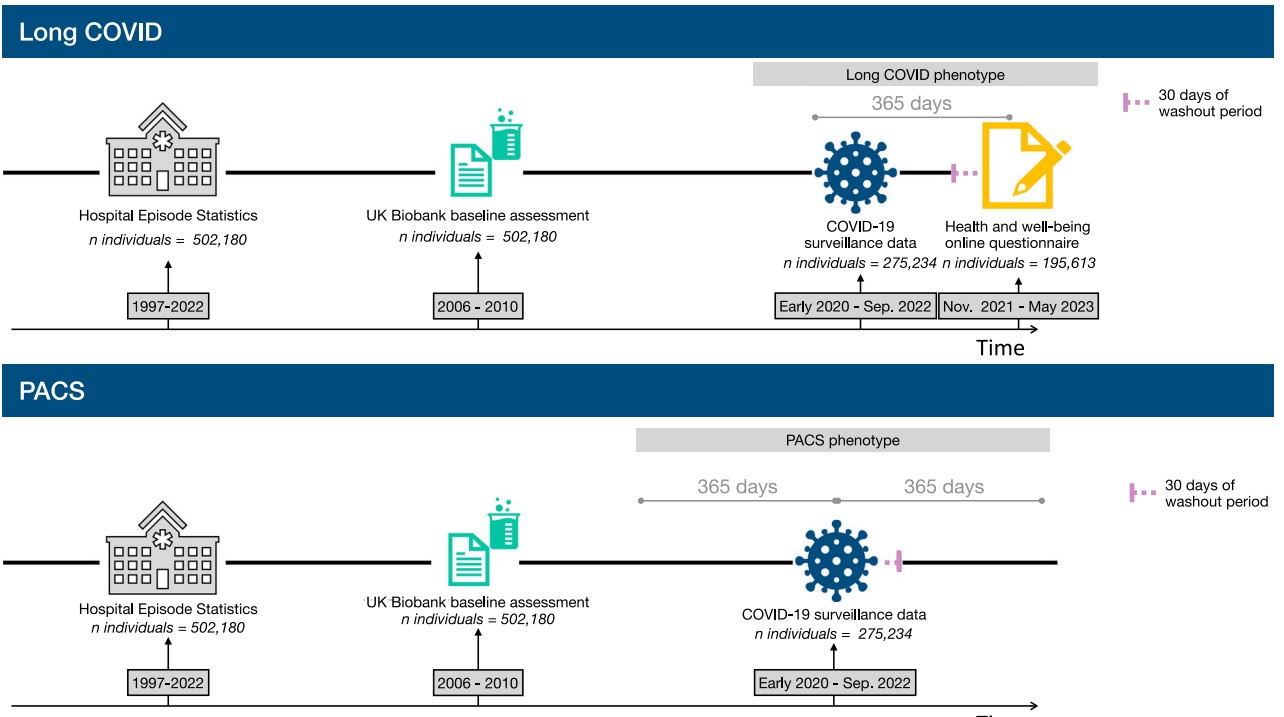

**Fig. 1 | Scheme of the overall study design to create the cohorts. Note:** PACS post acute COVID-19 sequelae, UK United Kingdom.

(OR$_{adjusted}$ = 1.20, 95% CI = 1.02 to 1.41; OR$_{adjusted}$ = 1.41, 95% CI = 1.22 to 1.63). Female sex was associated with an increased risk of LC (OR$_{adjusted}$ = 1.25, 95% CI = 1.14 to 1.35) (Fig. 3A).

Multiple biomarkers were associated with LC in the crude analyses. Higher levels of C-reactive protein and triglyceride were associated with a higher risk of LC (OR$_{crude}$ = 1.35, 95% CI = 1.17 to 1.55 per quintile 5; OR$_{crude}$ = 1.15, 95% CI = 1.10 to 1.21). Conversely, higher levels

of HDL-cholesterol were found to be associated with a lowered risk (OR$_{crude}$ = 0.73, 95% CI = 0.63 to 0.86 for quintile 5). Higher levels of IGF-1 and SHBG also appeared associated with a decreased risk of LC (OR$_{crude}$ = 0.92, 95% CI = 0.88 to 0.97; OR$_{crude}$ = 0.92, 95% CI = 0.87 to 0.97), whereas higher levels of hbA1c, cystatin C, urate and alanine aminotransferase were associated with increased risks: OR$_{crude}$ = 1.1, 95% CI = 1.04 to 1.17; OR$_{crude}$ = 1.11, 95% CI = 1.04 to 1.17; OR$_{crude}$ = 1.14, 95% CI = 1.07 to 1.2; and OR$_{crude}$ = 1.07, 95% CI = 1.02 to 1.13, respectively.

Most of these associations were attenuated and no longer significant after multivariable adjustment. However, higher levels of HDL-cholesterol remained associated with a decreased risk of LC (OR$_{adjusted}$ = 0.83, 95% CI = 0.70 to 0.98 for quintile 4), as were higher levels of IGF-1 (OR$_{adjusted}$ = 0.93, 95% CI = 0.88 to 0.98). Conversely, high levels of triglycerides and of vitamin D remained associated with an increased risk of LC: OR$_{adjusted}$ = 1.08, 95% CI = 1.01 to 1.15; OR$_{adjusted}$ = 1.05, 95% CI = 1.00 to 1.11, respectively (see Fig. 3B).

For pre-existing comorbidities, chronic kidney disease (CKD) and COPD were significantly associated with a higher risk of LC (OR$_{adjusted}$ = 1.48, 95% CI = 1.11 to 1.97; OR$_{adjusted}$ = 1.29, 95% CI = 1.08 to 1.54) (see Fig. 3C). Conversely, metastatic cancer was associated with a lower risk of LC (OR$_{adjusted}$ = 0.49, 95% CI = 0.28 to 0.86).

Results for PACS are presented in Fig. 4. We did not find any evidence of multicollinearity with any of the studied covariates (Supplementary Table 4). Older age was associated with a higher risk of PACS: OR$_{adjusted}$ = 2.41, 95%CI = 1.70 to 3.42 for those aged ≥75 compared to those <55. Obesity, living in socio-economically deprived areas, and male sex were also linked to a higher risk of PACS: OR$_{adjusted}$ = 1.39, 95% CI = 1.19 to 1.62; OR$_{adjusted}$ = 1.36, 95% CI = 1.17 to 1.58; and OR$_{adjusted}$ = 1.40, 95% CI = 1.24 to 1.59, respectively. Additionally, smoking was associated with a higher risk of PACS (OR$_{adjusted}$ = 1.30, 95% CI = 1.11 to 1.51) (see Fig. 4A).

In the crude analyses, higher levels of C-reactive protein, lipoprotein (a) and triglyceride were associated with higher risks of PACS

## Table 1 | Sociodemographic characteristics of the study cohorts

| Risk factor | Long COVID | | PACS | |
|---|---|---|---|---|
| | Controls | Cases | Controls | Cases |
| N | 5917 | 2751 | 106,467 | 1940 |
| Sociodemographic factors | | | | |
| Age [Mean (SD)] | 66.56 (7.24) | 66.31 (7.53) | 67.3 (8.2) | 71.63 (7.71) |
| Sex (%) | | | | |
| Female | 3061 (51.73) | 1486 (54.02) | 59,547 (55.93) | 834 (42.99) |
| Male | 2856 (48.27) | 1265 (45.98) | 46,920 (44.07) | 1106 (57.01) |
| BMI [kg/m2] [Mean (SD)] | 25.85 (3.85) | 26.43 (4.15) | 27.28 (4.56) | 28.94 (4.85) |
| Index of multiple deprivation | 13.43 (10.71) | 14.83 (11.55) | 16.76 (13.17) | 19.22 (14.38) |
| Ethnic background (%) | | | | |
| White | 5698 (96.3) | 2628 (95.53) | 100,694 (94.58) | 1827 (94.18) |
| Non-white | 219 (3.7) | 123 (4.47) | 5,773 (5.42) | 113 (5.82) |
| Smoking status (%) | | | | |
| Never | 3899 (65.89) | 1713 (62.27) | 61,364 (57.64) | 918 (47.32) |
| Previous | 1676 (28.33) | 866 (31.48) | 35,975 (33.79) | 785 (40.46) |
| Current | 342 (5.78) | 172 (6.25) | 9128 (8.57) | 237 (12.22) |

Age is measured at the index date (positive COVID-19 result). BMI, index of multiple deprivation, and smoking status were measured at the UK Biobank baseline assessment.

## Table 2 | Previous comorbidities of the study cohort

| Risk factor | Long COVID | | PACS | |
|---|---|---|---|---|
| | Controls | Cases | Controls | Cases |
| N | 5917 | 2751 | 106,467 | 1940 |
| Comorbidities [Cases (%)] | | | | |
| Acquired immunodeficiency syndrome (AIDS) | 2 (0.03) | 1 (0.04) | 36 (0.03) | 0 (0) |
| Asthma | 338 (5.71) | 205 (7.45) | 10,438 (9.8) | 312 (16.08) |
| Cancer | 1356 (22.92) | 640 (23.26) | 29,673 (27.87) | 778 (40.1) |
| Cancer - metastatic | 63 (1.06) | 16 (0.58) | 2006 (1.88) | 83 (4.28) |
| Cerebrovascular disease | 121 (2.04) | 46 (1.67) | 4029 (3.78) | 188 (9.69) |
| Congestive heart failure | 50 (0.85) | 27 (0.98) | 1251 (1.18) | 176 (9.07) |
| Chronic kidney disease | 125 (2.11) | 90 (3.27) | 5544 (5.21) | 327 (16.86) |
| Chronic obstructive pulmonary disease (COPD) | 363 (6.13) | 226 (8.22) | 12,508 (11.75) | 456 (23.51) |
| Dementia | 2 (0.03) | 0 (0) | 1029 (0.97) | 58 (2.99) |
| Diabetes | 154 (2.6) | 92 (3.34) | 7008 (6.58) | 383 (19.74) |
| Diabetes - organ damage | 10 (0.17) | 12 (0.44) | 832 (0.78) | 62 (3.2) |
| Fracture | 412 (6.96) | 173 (6.29) | 7971 (7.49) | 208 (10.72) |
| Hemiplegia | 11 (0.19) | 8 (0.29) | 695 (0.65) | 47 (2.42) |
| Liver disease - mild | 65 (1.1) | 46 (1.67) | 2558 (2.4) | 109 (5.62) |
| Liver disease - moderate to severe | 31 (0.52) | 23 (0.84) | 1224 (1.15) | 57 (2.94) |
| Myocardial infarction | 104 (1.76) | 52 (1.89) | 3100 (2.91) | 285 (14.69) |
| Peptic ulcer | 345 (5.83) | 198 (7.2) | 9710 (9.12) | 390 (20.1) |
| Peripheral vascular disease | 52 (0.88) | 23 (0.84) | 1666 (1.56) | 121 (6.24) |
| Rheumatoid arthritis | 57 (0.96) | 41 (1.49) | 2889 (2.71) | 138 (7.11) |

Comorbidities were measured before the positive COVID-19 result.

**Table 3 | Biomarker levels in the study cohorts**

| Risk factor | Long COVID | | PACS | |
|---|---|---|---|---|
| | **Controls** | **Cases** | **Controls** | **Cases** |
| *N* | 5917 | 2751 | 106,467 | 1940 |
| *Biomarkers [Mean (SD)]* | | | | |
| Alanine aminostransferase | −0.1 (0.94) | −0.06 (0.99) | −0.01 (1.01) | 0.1 (1.04) |
| Albumin | 0.17 (0.99) | 0.13 (0.99) | 0.05 (0.99) | −0.1 (1.02) |
| Alkaline phosphatase | −0.22 (0.93) | −0.19 (0.91) | −0.1 (0.97) | 0.13 (1) |
| Apolipoprotein A | 0.08 (0.96) | 0.04 (1) | −0.01 (0.98) | −0.16 (0.97) |
| Apolipoprotein B | −0.08 (0.94) | −0.03 (0.97) | −0.02 (0.97) | −0.08 (1.05) |
| Aspartate aminotransferase | −0.1 (0.9) | −0.09 (0.94) | −0.05 (0.98) | 0.09 (1.06) |
| C-reactive protein | −0.22 (0.79) | −0.15 (0.86) | −0.05 (0.95) | 0.21 (1.18) |
| Calcium | −0.01 (0.98) | 0.01 (0.97) | −0.02 (0.99) | −0.01 (1.02) |
| Cholesterol | −0.02 (0.92) | 0.01 (0.95) | −0.02 (0.96) | −0.18 (1.08) |
| Creatinine | 0.01 (0.94) | −0.01 (0.94) | −0.03 (0.96) | 0.2 (1.11) |
| Cystatin C | −0.33 (0.82) | −0.29 (0.85) | −0.17 (0.92) | 0.31 (1.1) |
| Direct bilirubin | 0.07 (1.02) | 0.07 (1.04) | 0.01 (1.01) | 0.07 (1.01) |
| Direct low-density lipoprotein | −0.03 (0.93) | 0 (0.95) | −0.01 (0.96) | −0.14 (1.06) |
| Gamma glutamyltransferase | −0.16 (0.85) | −0.11 (0.92) | −0.06 (0.95) | 0.18 (1.08) |
| Glucose | −0.15 (0.82) | −0.11 (0.84) | −0.08 (0.92) | 0.19 (1.2) |
| HbA1c1c | −0.27 (0.79) | −0.22 (0.83) | −0.13 (0.93) | 0.28 (1.14) |
| HDL cholesterol | 0.14 (0.98) | 0.08 (1.01) | 0.01 (0.98) | −0.22 (0.95) |
| IGF-1 | 0.25 (0.95) | 0.19 (0.97) | 0.1 (0.99) | −0.1 (1.02) |
| Lipoprotein (a) | −0.02 (0.98) | 0.02 (1.02) | −0.01 (1) | 0.05 (1.05) |
| Phosphate | 0 (0.98) | 0.01 (0.98) | 0.01 (1) | −0.01 (1) |
| SHBG | 0.06 (1.03) | 0.02 (1.01) | −0.01 (1.02) | −0.19 (0.89) |
| Testosterone | 0.07 (1.02) | 0.02 (1.01) | −0.03 (0.99) | 0.16 (0.98) |
| Total bilirubin | 0.1 (1.02) | 0.09 (1.05) | 0.02 (1.01) | 0.02 (0.99) |
| Total protein | 0.01 (0.99) | 0.01 (0.99) | 0 (0.99) | −0.03 (1.02) |
| Triglycerides | −0.2 (0.92) | −0.1 (1) | −0.05 (0.99) | 0.16 (1.07) |
| Urate | −0.12 (0.97) | −0.07 (1.01) | −0.05 (0.99) | 0.28 (1.02) |
| Urea | −0.11 (0.92) | −0.12 (0.92) | −0.08 (0.96) | 0.16 (1.07) |
| Vitamin D | 0.07 (0.97) | 0.08 (0.97) | 0.02 (0.99) | 0.03 (1.02) |

Biomarkers were measured at the UK Biobank baseline assessment. Notice that biomarkers levels have been standardised using z-scores.

(OR$_{crude}$ = 2.04, 95% CI = 1.76 to 2.37 for quintile 5 (Fig. 4B); OR$_{crude}$ = 1.05, 95% CI = 1.00 to 1.1; OR$_{crude}$ = 1.33, 95% CI = 1.15 to 1.55 for quintile 5, respectively). Conversely, elevated levels of LDL and HDL-cholesterol were found to be associated with a reduced risk (OR$_{crude}$ = 0.69, 95% CI = 0.60 to 0.79 for quintile 5; OR$_{crude}$ = 0.57, 95% CI = 0.49 to 0.67 for quintile 5). Higher alkaline phosphatase levels were also associated with an increased PACS risk (OR$_{crude}$ = 1.8, 95% CI = 1.54 to 2.1 for quintile 5), whereas high levels of calcium and vitamin D were associated with a decreased risk (OR$_{crude}$ = 0.86, 95% CI = 0.75 to 1 for quintile 4; OR$_{crude}$ = 0.95, 95% CI = 0.9 to 0.99). IGF-1 and SHBG were associated with a lower risk of PACS (OR$_{crude}$ = 0.72, 95% CI = 0.62 to 0.83 for quintile 5; OR$_{crude}$ = 0.88, 95% CI = 0.83 to 0.93). Finally, high levels of HbA1c, cystatin C, and urate were also found to increase PACS risk (OR$_{crude}$ = 2.04, 95% CI = 1.74 to 2.38 for quintile 5; OR$_{crude}$ = 1.35, 95% CI = 1.29 to 1.42; OR$_{crude}$ = 1.70, 95% CI = 1.43 to 2.03 for quintile 5).

Many associations were attenuated after multivariable adjustment. Here, we summarise the associations that remained significant. Higher levels of alkaline phosphatase, HbA1c, and cystatin C remained statistically significantly associated with an increased risk of PACS after adjusting for other variables (OR$_{adjusted}$ = 1.35, 95% CI = 1.14 to 1.59 for quintile 5; OR$_{adjusted}$ = 1.29, 95% CI = 1.09 to 1.54 for quintile 5; OR$_{adjusted}$ = 1.09, 95% CI = 1.03 to 1.15). Higher levels of IGF-1 remained associated with a decreased risk of PACS: OR$_{adjusted}$ = 0.84, 95% CI = 0.72 to 0.98 (see Fig. 4B).

Almost all the comorbidities explored were associated with an increased risk of PACS, except cerebrovascular disease, dementia, fracture, hemiplegia and liver disease (see Fig. 4C).

## Sensitivity analysis

After removing liver transaminases and diabetes biomarkers (glucose, IGF1, and HbA1c) from the analysis, our adjusted results remained consistent with the main analyses (see Supplementary Figs. 3 and 4).

We also conducted a sex-stratified analysis to examine the associations of SHBG and testosterone with the respective study outcomes. However, as testosterone was excluded from the analysis because it was highly correlated (|*r*|>0.5) with creatinine, we focus solely on SHBG. In these sex-stratified analyses, higher SHBG was associated with a decreased risk of LC and PACS in females only (Supplementary Figs. 5 and 6).

We modified the definition of LC cases to minimise the risk of misclassification. We included only participants reporting at least three World Health Organisation (WHO) LC symptoms at the time of answering the online questionnaire. Those reporting two or fewer symptoms were classified as LC controls. This resulted in a set of 594 LC cases (3%) and 8,079 LC controls (97%). The associations observed

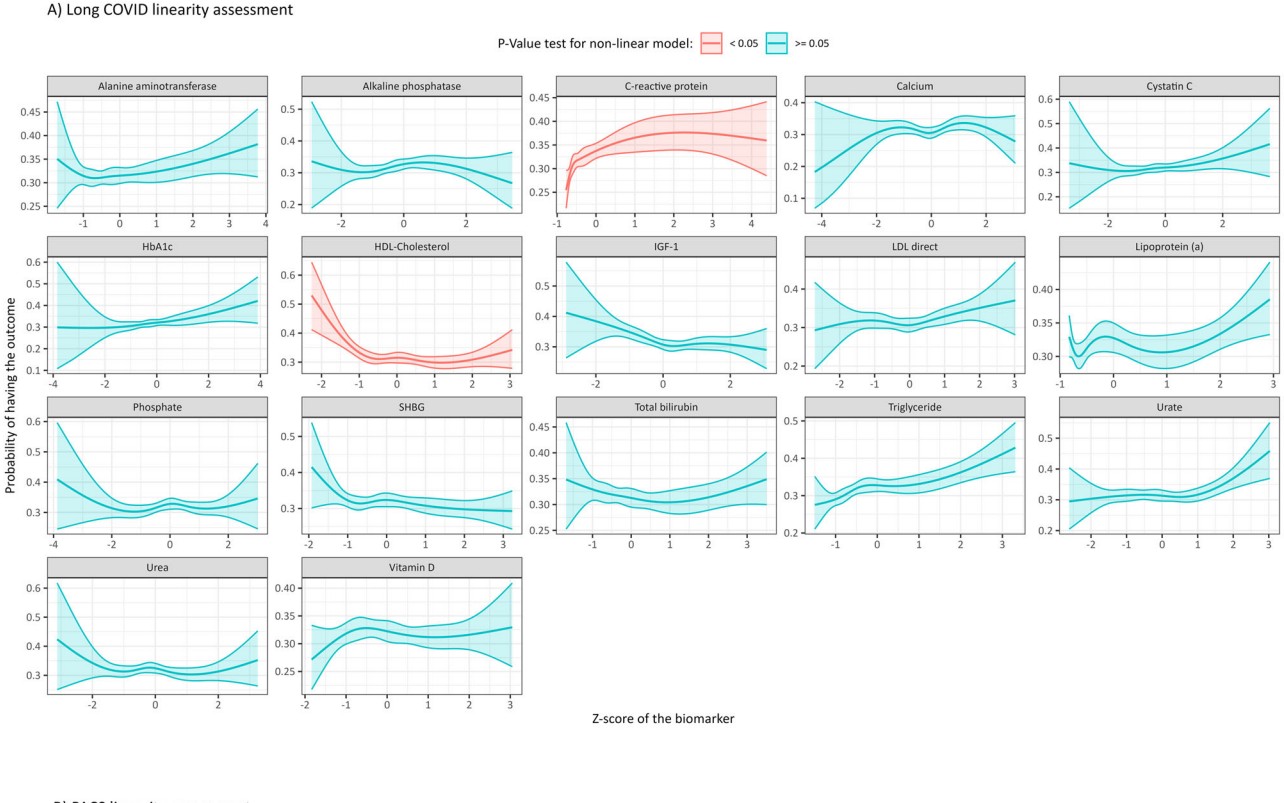

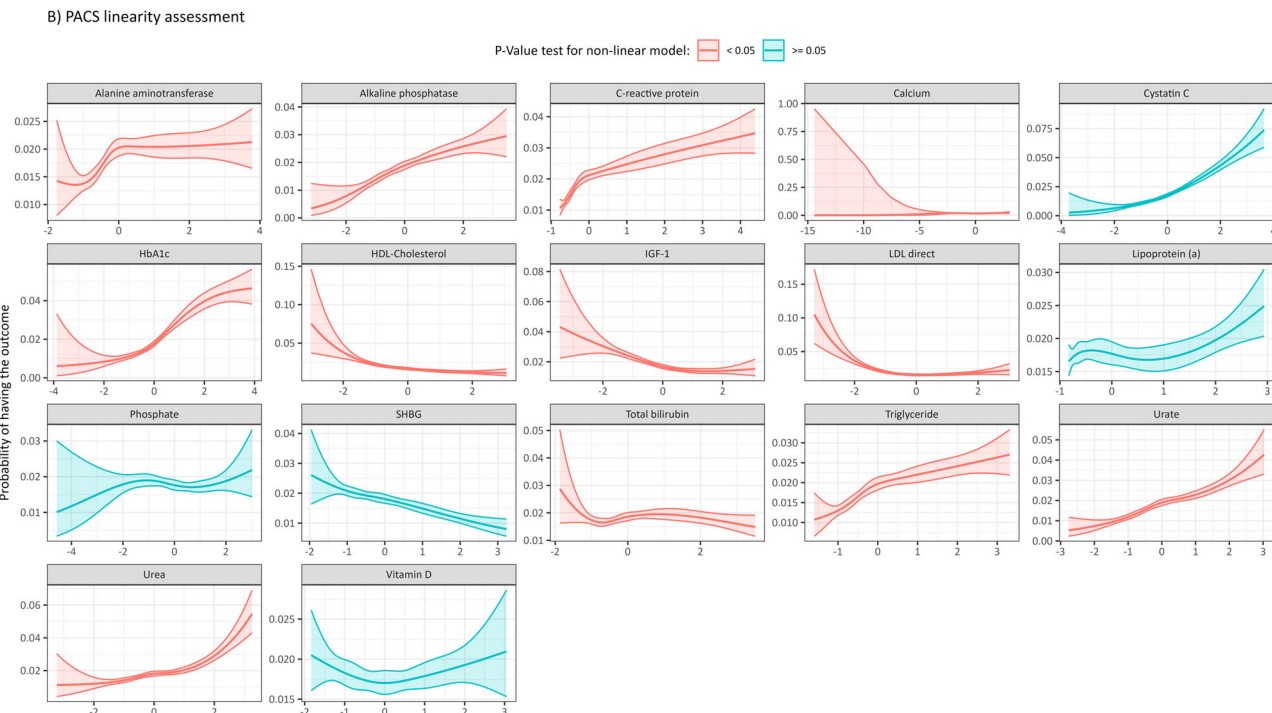

**Fig. 2 | Linearity assessment.** Evaluation of the non-linear relationship between the biomarkers and **A** Long COVID or **B** PACS using a natural cubic spline model. Error bands were calculated using 95% confidence intervals. The two-sided *P*-value was calculated using ANOVA. Red colour indicates a *P*-value smaller than 0.05, indicating that the non-linear model is a better fit compared to the linear model. Blue colour indicates a *P*-value higher than 0.05, suggesting that the non-linear model and the linear model are similar. *Note*: Notice that the non-linear test was performed between the log-odds of the outcome model, not the probability of the outcome itself. However, the probability of the outcome is plotted as it is more intuitive and accessible for interpretation. Notice that the shape of the curve is similar when plotted on the log-odds scale.

in these analyses were consistent with those from the main results (see Supplementary Fig. 7), although with an increase in the confidence interval values. Additionally, we also modified the LC definition to include participants reporting symptoms after 3 months (90 days) of the infection. This yielded a cohort of 2383 cases (31%) and 5286 controls (69%). The direction and association observed were consistent with those from the main results (see Supplementary Fig. 8).

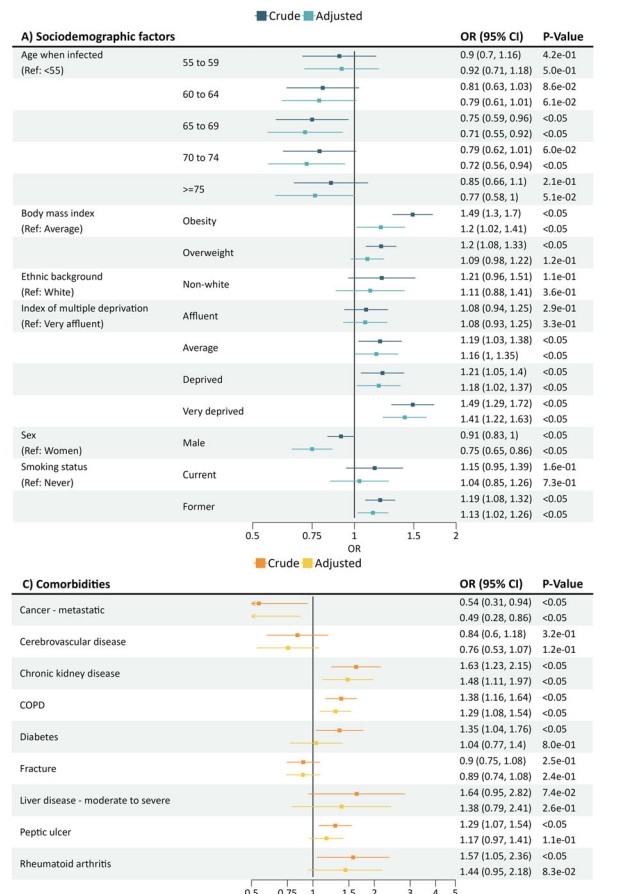

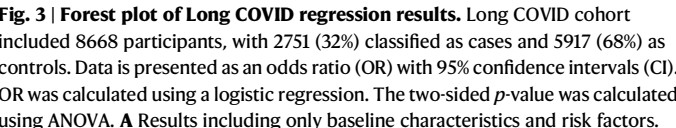

**Fig. 3 | Forest plot of Long COVID regression results.** Long COVID cohort included 8668 participants, with 2751 (32%) classified as cases and 5917 (68%) as controls. Data is presented as an odds ratio (OR) with 95% confidence intervals (CI). OR was calculated using a logistic regression. The two-sided p-value was calculated using ANOVA. **A** Results including only baseline characteristics and risk factors. Dark blue corresponds to crude findings; light blue corresponds to adjusted findings. **B** Results including only biomarker results. Dark maroon corresponds to crude findings; light maroon corresponds to adjusted findings. **C** Results including only the comorbidities results. Orange corresponds to crude findings; yellow corresponds to adjusted findings.

Finally, after adjusting for multiple testing, LC results still showed a statistically significant association with socio-economic deprivation and sex (see Supplementary Fig. 9). For PACS, the results adjusted for multiple testing showed a positive statistically significant association with older age, obesity, deprivation, and sex. Higher levels of alkaline phosphatase remained statistically significantly associated with an increased risk of PACS. The results for comorbidities were consistent with those observed in the main analysis (see Supplementary Fig. 10).

## Discussion

We conducted a data-driven hypothesis-free analysis to explore the association of 55 prespecified candidate determinants, including 6 socio-demographics, 30 clinical biomarkers, and 19 comorbidities, with the risk of LC and PACS, separately.

Among the socio-demographic factors, we found younger age (<55), obesity, deprivation, female sex and smoking associated with an increased risk of LC. Regarding the biomarkers, elevated levels of cardiovascular biomarkers of C-reactive protein, lipoprotein (a), and triglycerides were linked to a higher LC risk, whereas higher HDL-cholesterol levels were associated with a decreased risk. Additionally, higher levels of cystatin C, urate, HbA1c, vitamin D, and alanine aminotransferase were found to be linked to higher LC risk. Finally, higher levels of IGF-1 and SHBG were associated with reduced risk, although the association with SHBG was only observed in females. When it comes to comorbidities, a history of CKD or COPD prior to COVID-19 infection was also associated with an increased risk. The direction of

these associations remained despite applying a stricter definition of LC (requiring at least three self-reported symptoms) in the sensitivity analyses, except for lipoprotein (a).

For the PACS outcome, older age, obesity, deprivation, male sex, and smoking were associated with an increased risk. Most cardiovascular and inflammation biomarkers, including C-reactive protein, LDL, HDL-cholesterol, lipoprotein (a), and triglycerides, all showed an association with a higher risk of PACS. Additional biomarkers associated with a higher PACS comprised HbA1c, alkaline phosphatase, cystatin C, urate, calcium, or vitamin D. Similar to LC, IGF-1 and SHBG were associated with a decreased risk of PACS, with the latter only in females. A history of several comorbidities, such as cancer, congestive heart failure, CKD, and others, was also found to increase the risk of PACS.

Despite the retrospective design of our study, most of our findings are consistent with previous evidence[7], supporting female sex, obesity, and socioeconomic deprivation as key socio-demographic factors associated with LC. In addition, prior studies have also reported that smoking or having a Black or Hispanic ethnic background is associated with an increased risk[10]. Our research also verified both of these, albeit with nominal statistical significance.

Notably, although old age has been an established risk factor for severe COVID-19 in the acute phase, its role in long-term outcomes remains unclear. A systematic review and meta-analysis of nine studies found that adults older than 40 had a higher risk of LC compared to younger adults[7]. However, our study found that ageing was only

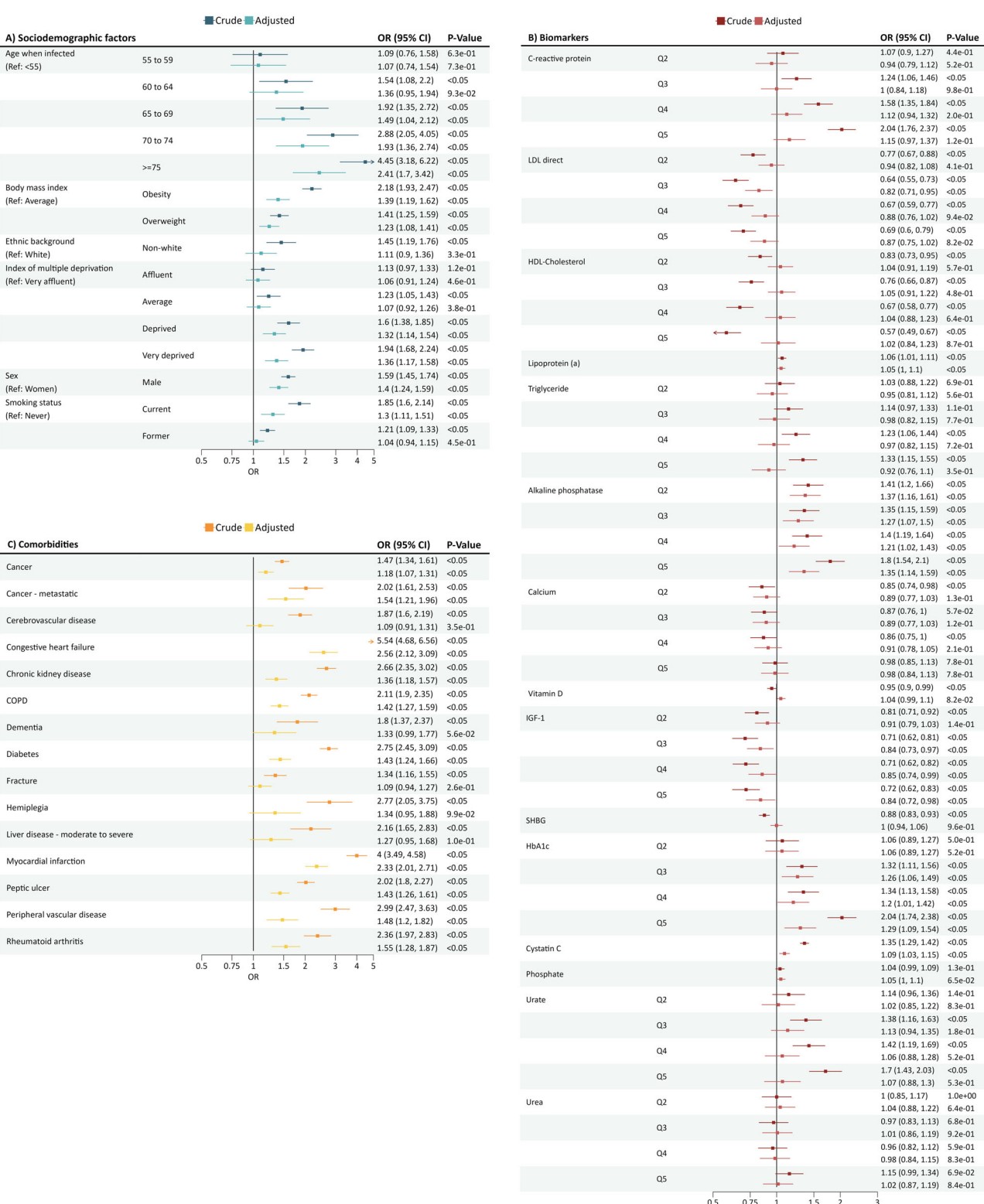

**Fig. 4 | Forest plot of PACS regression results.** PACS cohort included 108,407 participants, with 1940 (2%) classified as cases and 106,467 classified as controls (98%). Data is presented as an odds ratio (OR) with 95% confidence intervals (CI). OR was calculated using a logistic regression. The two-sided *p*-value was calculated using ANOVA. **A** Results including only baseline characteristics and risk factors. Dark blue corresponds to crude findings; light blue corresponds to adjusted findings. **B** Results including only biomarker results. Dark maroon corresponds to crude findings; light maroon corresponds to adjusted findings. **C** Results including only the comorbidities results. Orange corresponds to crude findings; yellow corresponds to adjusted findings.

positively associated with the PACS phenotype. In contrast, it was negatively associated with LC. This is in line with our previous research[11] and other studies[12]. One possible explanation for this discrepancy could be either differential mortality rates in older groups compared to younger ones or differences in defining LC phenotypes; most prior studies relied on clinical diagnosis from primary or secondary care data, whereas our study used patient self-reported outcomes. Reporting bias may also play a role, as older individuals with more severe outcomes may have been less likely to complete the health and well-being questionnaire. Importantly, we found females were more prone to LC and males to PACS. Similar results have been seen in previous studies, where the likelihood of developing long COVID syndrome was significantly higher in females compared to males[13], although endocrine and renal disorders were higher among males. Generally, females mount stronger innate and adaptive immune responses to infections and vaccinations compared to males[14,15]. This can be beneficial in clearing acute infections, but it may also predispose females to prolonged immune dysregulation and a higher risk of developing persistent inflammation or even autoimmune conditions. Instead, males are more likely to experience severe acute COVID-19, with higher rates of hospitalisation, ICU admission, and death compared to females. This increased severity during the acute phase can lead to more significant organ damage, which predisposes more males to PACS.

Regarding biomarker findings, inflammatory biomarkers have been most frequently reported to be elevated in acute severe COVID-19[16]. Our study suggests that baseline inflammation, measured by C-reactive protein levels in plasma before COVID-19 infection, is associated with an increased risk of LC. Importantly, two cardiometabolic biomarkers (low HDL and high triglycerides) were also associated with an increased risk of LC in our study, corroborating previous reports[17,18]. We also observed that most pre-existing comorbidities are risk factors for the development of LC after the infection. For example, CKD and COPD were the two conditions most strongly associated with LC in this study. Similar results were found in a prospective UK cohort study, where COPD and benign prostatic hyperplasia showed the strongest associations with an increased risk of LC symptoms[10].

Reassuringly, most factors linked with LC also appeared to impact the PACS in a similar direction, except for age and gender. This finding is important to highlight the similarities and differences of two types of traits occurring during the post-acute stage of COVID-19[19].

Our study focused on identifying determinants of LC and PACS, but it is important to consider the broader context of interventions that may influence these outcomes. For example, vaccination before infection has also been reported to partially mediate the risk of sequelae at 6 months by 15%[20]. A recent study that conducted a multinational staggered cohort study[21] suggested that vaccination could mitigate the risk of acute and subacute post-COVID-19 venous thromboembolism and heart failure by 40% and 30%, respectively, for 90–180 days post-COVID-19 infection. However, we did not have data regarding vaccination, so we could not explore how it affects the outcomes.

Additionally, the use of antivirals (molnupiravir or nirmatrelvir) within 5 days after infection has also been associated with a lower risk of sequelae by 14% and 26%, respectively[22]. However, only patients at risk of progression to severe COVID-19 are eligible for these drug interventions during the acute phase, and their risk-benefit in a wider population with milder infection, as well as the post-acute stage of disease, remains unclear. Importantly, while most post-COVID symptoms are more common following severe cases, the highest overall burden is still among those with a history of mild-to-moderate COVID-19 illness, as they represent most infected individuals now and likely in the future. Our findings on potential biomarkers in general support the promise of drug development targeting inflammation, the immune system, or blood clots-pathways currently under clinical

investigation[23]. However, the efficacy of these approaches remains unpredictable due to the interplay and complexity of the underlying mechanisms[4,24,25].

This study has several strengths that enhance the reliability of its findings. First, we used unique linked UK Biobank data, which included extensive biomarker measurements, sociodemographic factors and comorbidity information. This comprehensive dataset enabled us to investigate a wide range of variables associated with both LC and PACS. Additionally, we used a study specifically designed to detect LC symptoms within the UK Biobank cohort, providing a robust phenotype for LC. Second, we separately examined LC and PACS, being, to our knowledge, the first study to date to explore the key variables associated with these two outcomes independently. Importantly, we identified key differences in the associations between the variables and these outcomes.

The limitations of the study must be considered when interpreting the findings. First, our analysis describes associations, not causal effects. Further research is needed to establish the nature of these associations. Second, UK Biobank participants are known to be healthier and older than the general population, which limits the generalisability of our results. Third, some sociodemographic data and all biomarker' measurements were collected at the UK Biobank's first assessment, and may have changed over time, potentially introducing bias. However, the time span between measurement and outcome is likely attenuating associations due to random misclassification. Fourth, most of the participants recruited in the UK Biobank are White, and other ethnic backgrounds, such as Asian or Asian British, Black or Black British, or Chinese, may be underrepresented. Fifth, patient-reported outcomes (PROs) are inherently subjective, which may result in misclassification of LC cases due to variability in symptom reporting. Additionally, the self-reported nature of the questionnaire can introduce recall bias, as participants may not accurately remember the symptoms or their length. Sixth, the definition of PACS relies on electronic medical records captured during hospitalisation, likely leading to an underestimation of patients with milder complications. Finally, we did not have data on participants' vaccination status. Vaccination has been previously linked to a reduced risk of both LC and PACS and may have an impact on our results[21,26].

This study reports differences in the risk factors contributing to LC and PACS, with young age and female sex identified as risk factors for LC, while male sex and older age are associated with an increased risk of PACS. We also report novel biomarkers associated with LC and PACS. Specifically, results suggest an association between inflammation, cardiovascular, and liver function markers and the risk of LC. Conversely, higher levels of IGF-1 and SHBG were associated with a decreased risk of LC. Findings for PACS are similar, with the addition of increased risks associated with high levels of HbA1c, cystatin C, or urate. These results have significant potential to inform clinical and public health practices to identify high-risk groups for LC and PACS, with the aim of targeting interventions, including booster vaccination. This study will also inform future research on potential drug targets for the prevention of LC or PACS.

## Methods
### Data sources
UK Biobank is a large-scale, population-based prospective cohort of over 500,000 individuals aged 40–69 years on recruitment (2006–2010)[27]. It contains detailed information on sociodemographic, lifestyle factors, biomarkers, and a range of patient-reported outcomes. Follow-up is primarily conducted through linkage to electronic health records from primary and secondary care. In the present study, we used Hospital Episodes Statistics (HES), which includes diagnostic and procedure data from 1997 until October 2022 for all participants resident in England. In addition, the Public Health England's Second-Generation Surveillance System was linked to UKBB[28], providing

additional information on PCR-confirmed COVID-19 infection status for UKBB participants. This latter linkage included three different datasets: one covering England (with available data from early 2020 until September 2022), Scotland (2020–November 2022), and Wales (2020–December 2022).

Additionally, an online survey was performed among UK Biobank participants to actively collect patient-reported data on health and well-being during the pandemic. In this survey, 201,684 participants completed the questionnaire between June 2022 to May 2023, including 45 questions related to COVID-19 symptoms. More information on each question is provided in Supplementary Table 5.

## Study design
We designed two case-control studies nested within a cohort of UK Biobank participants infected with COVID-19 during the period from 2020 until the end of 2022. Figure 1 and Supplementary Figs. 11 and 12 illustrate the study design.

## Covariates and outcomes
**Definition of Long COVID.** We first curated a base COVID-19 infection cohort including all participants who completed the Health and well-being survey (around 200,000), had a valid linkage to COVID-19 surveillance data, and with a positive COVID-19 PCR test result 1 year to 30 days before completing the survey. Only the most recent infection closest to the survey completion data was considered for the analysis (see Supplementary Fig. 11).

Among the COVID-19 infection cohort, we identified LC cases based on the WHO Delphi consensus definition[29]. Specifically, we mapped the available symptoms from the Health and Well-Being survey to the WHO definition (Supplementary Table 6). Participants who did not answer or preferred not to answer any of the questions regarding symptoms were excluded. Participants reporting pre-existing symptoms were also excluded (see Supplementary Note 2 for more details on how pre-existing symptoms were defined). The infected participants who had one or more WHO-listed LC symptoms beyond the acute period (30 days) were classified as cases, whereas others were included as controls. Notice that our definition differs slightly from that of the WHO, which defines LC as symptoms persisting beyond 90 days post-infection. At the time this study was conducted, there was no universally accepted definition. Previous studies have used a 30-day threshold, and current NHS guidance advises individuals to consult a GP if symptoms persist beyond 30 days[30].

**Definition of PACS.** Similarly to the above, we generated a COVID-19 infection cohort for the sampling of PACS cases, now including all UK Biobank participants with a positive COVID-19 test result. Participants with a PACS diagnosis one year before or within 30 days after the infection date were excluded (see Supplementary Fig. 12). Cohort participants with a PACS diagnosis recorded between 30 days and 1 year post-infection were classified as cases, whereas all other participants in the COVID-19 cohort were classified as controls. In cases of multiple COVID-19 infections, only the earliest infection was considered for analysis. The diagnoses associated with PACS outcome were selected based on clinical knowledge and prior literature[17,18,31–38]. Supplementary Table 7 provides a list of ICD-10 codes used in this study[11,39].

**Candidate covariates.** Several socio-demographic characteristics (6), clinical biomarkers (30) and comorbidities (19) were pre-specified for the analysis of associations with LC and PACS and modelled separately. Socio-demographic characteristics and biomarkers were extracted from the baseline UK Biobank assessment (conducted between 2006 and 2010), where information was collected through touchscreen questionnaires and biological samples. Biomarkers were selected for the analysis based on their availability in the UK Biobank. The UK Biobank selected these biomarkers because they represent established risk factors for disease and are clinical diagnostic measures. Comorbidities were selected based on the Charlson Comorbidity Index[40] and phenotyped using linked HES data.

## Statistical analysis
**Exploratory data analysis.** We performed a descriptive analysis for each of the covariates of interest to understand the extent of missing data, examine the distribution, and identify outliers. Variables with a value of *"I don't know"* or *"I prefer not to answer"* were considered missing. Variables with more than 50% of missing data were excluded from further analysis. Multiple imputation was used for variables with less than 50% of missing data. For continuous variables, we calculated the mean, standard deviation, quantiles, and their distribution. For categorical variables, we report proportions and depict the data using histograms.

Data curation and correlation analysis were performed before the main analysis. See Supplementary Note 3 for more information.

**Linearity assessment.** We explored potential non-linearities between each biomarker and both study outcomes using natural cubic spline curves (adjusted using 3 knots). We compared the natural cubic spline model to the logistic regression model, and used ANOVA to assess whether there was a statistically significant difference between the two models. A $p$-value < 0.05 was used to indicate that the non-linear model provided a better fit to the data, suggesting a potential non-linear association between the biomarker and the logit (log-odd) function of the study outcome.

**Variable selection.** We used logistic regression with LASSO penalisation to select key variables for modelling. Three separate regressions were performed, one for each group of covariates, with the aim of identifying as many risk factors as possible: sociodemographic, biomarkers, and comorbidities. The biomarker and comorbidity models were further adjusted for age and sex. We used k-fold cross-validation to tune the lambda parameter, choosing the value that minimised the cross-validated error (calculated based on squared error for Gaussian models).

Variables that showed a non-linear relationship with the study outcomes were included in the LASSO model using spline functions.

**Outcome modelling.** Covariates selected from the LASSO were included in a logistic regression model to estimate associations with LC and PACS, respectively. To improve model interpretability, body mass index, age when infected by COVID-19, IMD and biomarkers showing a potential non-linear relationship were categorised. Details of the specific criteria used to categorise the variables can be found in Supplementary Note 4.

We performed two regression model analyses. First, in what we called a crude analysis, we modelled each variable separately, adjusting only for age (as a continuous variable) and sex. The second model (adjusted analysis) included all variables simultaneously, excluding those with a variance inflation factor (VIF) greater than 5 to minimise collinearity and overfitting.

## Sensitivity analyses
First, we conducted a sensitivity analysis, removing the biomarkers known to be associated with specific comorbidities from the multi-variable adjusted regression model. These include HbA1c, glucose and IGF-1 with diabetes and alanine aminotransferase, alkaline phosphatase, aspartate aminotransferase and gamma glutamyltransferase with liver disease.

Second, analyses of testosterone and SHBG were stratified by sex, given their known association with sex.

Thirdly, we repeated the main analysis using an alternative LC definition where participants were classified as cases if they reported at least three WHO proposed symptoms. Given the reliance on self-reported data, this approach aimed to reduce the risk of mis-classification. Those with fewer than two symptoms or who did not report any symptoms were classified as controls.

Fourth, we repeated the main analysis using the WHO definition for LC, where participants were classified as cases if they had the symptoms after three months (90 days) post-infection.

Finally, we applied Bonferroni corrections for multiple comparisons, and adjusted p-value thresholds and 95% confidence intervals accordingly. The number of comparisons was determined by the number of variables selected by the LASSO in each subgroup (socio-demographics, biomarkers, and comorbidities).

## Software and implementation
Data manipulation was done with R software (version 4.3.0), and the main packages used for analysis and reporting include mice[41], dplyr[42] (version 1.1.3), ggplot2[43] (version 3.5.1), and glmnet[44,45] (version 4.1.8).

## Reporting summary
Further information on research design is available in the Nature Portfolio Reporting Summary linked to this article.

## Data availability
The UK Biobank patient-level data are available under restricted access for bona fide researchers; access can be obtained by applying at http://ukbiobank.ac.uk/register-apply/. Raw data are protected and are not available due to data privacy laws. All participants provided informed written consent to take part in the study. Ethics approval for the UK Biobank was granted by the North West Multi-Centre Research Ethics Committee in 2006 and was updated regularly after that (https://www.ukbiobank.ac.uk/learn-more-about-uk-biobank/about-us/ethics). This study was conducted after approval by the UK Biobank under application reference 151425.

## Code availability
All the analytical code is publicly available in GitHub[46], which is in line with current recommendations to increase transparency and reproducibility[47].

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

## Acknowledgements

This study was conducted after approval by the UK Biobank under application reference 151425. This work uses data provided by patients and collected by the NHS as part of their care and support. This research used data assets made available by National Safe Haven as part of the Data and Connectivity National Core Study, led by Health Data Research UK in partnership with the Office for National Statistics and funded by UK Research and Innovation (research which commenced between 1st October 2020–31st March 2021 grant ref MC_PC_20029; 1st April 2021–30th September 2022 grant ref MC_PC_20058). The research was supported by the National Institute for Health and Care Research (NIHR) Oxford Biomedical Research Centre (BRC) and by Gilead Sciences, Inc. DPA is funded through a NIHR Senior Research Fellowship (Grant number SRF-2018-11-ST2-004). The views expressed in this publication are those of the author(s) and not necessarily those of the NHS, the National Institute for Health and Care Research or the Department of Health.

## Author contributions

M.A.H., S.I., D.P.A., and J.X. drafted the study protocol. M.A.H., M.C., J.X. performed the statistical analysis. M.A.H., S.I., J.W., Y.L., W.N., M.B., D.P.A., and J.X. interpreted the data and the study findings. M.A.H. and J.X. drafted the study paper. S.I., J.W., Y.L., W.N., M.B., M.C., and D.P.A. reviewed the study paper. All authors have approved the submitted version.

## Competing interests

DPA research group from the University of Oxford has received research grants from the European Medicines Agency, from the Innovative Medicines Initiative, from Gilead Science, and from UCB Biopharma. S.I., J.J.W., W.N., Y.L., and M.B. are employees of Gilead Sciences and may own stock in the company. The remaining authors declare no competing interests.
