## [Transparent Peer Review file · Nature Communications]

Sociodemographic factors, biomarkers and comorbidities associated with post-acute COVID-19 sequelae in UK Biobank

Corresponding Author: Professor Daniel Prieto-Alhambra

Version 0:

Reviewer comments:

Reviewer #1

(Remarks to the Author)

This case-control study used UK Biobank data to examine clinical factors associated with Long COVID. A range of sociodemographic factors, biomarkers, and comorbidities were included. Strengths of the study included the incorporation of both self-reported and hospital-recorded symptoms, as well as the extensive biomarker data available in the Biobank data. The data are carefully curated, and the manuscript is well written. However, I have the following concerns.

1. The authors classified Post-COVID-19 Conditions (PCC) into two main categories: Long COVID and post-acute complications of SARS-CoV-2 infection (PACS) to distinguish between self-reported and more severe thromboembolic or cardiovascular complications. Specifically, Long COVID is defined as symptoms reported by patients themselves beyond 30 days, which differs from how Long-COVID is defined by the WHO and many other studies. While the operational definition of Long COVID do vary across studies, it would be helpful to briefly review how long-COVID has been previously defined and measured in the literature, given the growing body of research on this topic.
2. Only two UK studies were discussed in the Introduction as prior evidence. It would be more informative to summarize existing evidence from a higher level by including findings from systematic review and meta-analysis (some of which the authors have already cited).
3. PPC typically includes both the continuation of initial symptoms and the emergence of new symptoms following the initial infection. The authors included individuals with PACS diagnosis recorded between 30 days and 1-year post-infection, excluding those with pre-existing symptoms and those with diagnosis within the first 30 days. The analytical sample is therefore presents a highly selective group - those who did not experience any symptoms within 30 days after the infection. Factors associated with developing new symptoms after 30 days post-infection may be substantially different from those associated with persistence of symptoms. The rationale for excluding people with persistent symptoms should be clarified.
4. The definition of Long COVID relies heavily on self-reported symptoms, with durations spanning from less than 2 weeks to over 12 weeks, which introduces potential recall bias. When a relative coarse measure of symptom duration is used in combination of a specific date of infection, the risk of misclassification is high and need to be discussed.
5. Self-reported measures may also be associated with psychological factors and the presence of neuropsychiatric and mood disorders. Only dementia was included among comorbidities in the analysis. It would be helpful for the authors to explain how comorbidities were selected.
6. A series of exploration data analysis has been performed, including linearity assessments as shown in Figure 1. However, in logistic regression, the assumption of linearity applies to the relationship between the explanatory variables and the logit (log-odds) of the outcome – not the probability itself. Since the y-axis in Figure 1 depicts the probability of having the outcome, the figure maybe misleading. It is also not clear how this test is performed. The authors should further clarify their approach and verify the validity of the test.
7. Variable selection was performed stepwise by fitting three LASSO regressions. What is the rationale for not including all

variables in a single regression as biomarkers may also be correlated with comorbidities? Is this because of a concern of power? The rationale for this approach should be explained more clearly.

(Remarks on code availability)

The code is clearly annotated. There is a README file, but it requires login to see the details. I am not able to run the code as I don't have access to the data, hence I cannot assess the reproducibility of the code.

Reviewer #2

(Remarks to the Author)

This is a case-control study utilizing COVID-19 infected individuals in the UK Biobank to explore the association of sociodemographic factors, clinical biomarkers, and comorbidities with the risk of Long COVID (LC) and Post-Acute COVID-19 Syndrome (PACS). The authors found an association between inflammation, cardiovascular, and liver function markers to be associated with risk of LC, and an addition of high levels of HbA1c, cystatin C, and urate to be association with PACS. This study benefits from the comprehensive statistical analyses and fruitful covariates to be included, while the summary of pre-existing evidence and result interpretation can be greatly improved. I do have several comments for the authors to consider:

1. It is suggested to provide a more comprehensive summary on post-covid19 conditions in the Introduction. The authors mentioned that risk of PCC was potentially independent of the severity of the acute COVID-19 infection. I however found some evidence showing risk of PCC was higher among individuals recovering from the most severe form of acute COVID-19 infection (Magnúsdóttir I, 2022; Shen Q, 2023).
2. It appears unclear to me how LC and PACS, the two primary outcomes, was defined by definitions. It seems there're overlapped symptoms in these two outcomes? I do not find evidence showing a clear definition or clear thresholds to separate these outcomes by checking the cited references. I do not follow why and how PACS refers to more severe thromboembolic or cardiovascular complications. Please provide justifications on why studying these two outcomes in this study is important, and what additional evidence could be provided in current study for LC in addition to existing studies.
3. I am not following the Suppl.Fig1 flowchart. What's the difference between "restrict to participants that answered the health and well-being questionnaire", and "restrict to participants that answered yes/no to ...questionnaire"? If this is to exclude missing value or prefer not to answer, it will be helpful to clarify numbers and reasons for this exclusion. The authors may consider to justify why restrict to the latest record for covid-19 infection for LC cohort while consider the earliest infection for PACS cohort? Suppl. Figure 11 LC cohort can be followed, but PACS cohort is very hard to understand. Suggest to add description or notes under this picture to help understanding.
4. I am confused by the definition of pre-existing symptoms. The authors mentioned "If participants reported options (1), (2), (4), (5), we assumed that the patient has not have persistent symptoms. If participants reported option (3) we assumed that the persistent symptoms had a length of four weeks (28 days). If participants reported option (4), we directly assumed that the participants have persistent symptoms." Where exactly option (4) participants were grouped to?
5. I am a little puzzled by the clinical implementation of this study. How can biomarkers and characteristics assessed over 10 years prior to COVID-19 infection being able to be used for prediction of LongCOVID syndrome, or help identify high risk groups?
6. The authors lack justifications on selection of risk factors or biomarkers. Why include these covariates and the supporting evidence?
7. The CDC has recommended a BMI less than 18.5 to be underweight and 18.5 to 25 to be "healthy weight". Any reason not to consider underweight in the categorization of BMI?
8. Variables with more than 50% of missing data were excluded from further analysis. How missing value was handled for those variables with less than 50% missing data?
9. The authors applied LASSO regression for modelling. It will be better to provide more details how variables were fitted in the model, and provide appropriate citations. For instance, why select one for each group of covariates? what parameters were used for spline function?
10. Why fit logistic regression models separately for the three categories of covariates? In sensitivity analyses, why exclude several biomarkers to be associated with some comorbidities, any reference or evidence? Using three symptoms in sensitivity analyses for LC definition should be justified.
11. In discussion, the authors mentioned an opposite trend of age to be associated with lower risk of LC, and explained that differential mortality rates could be the reason behind. The authors may also consider the reporting bias in questionnaire, where the most severe patients of older ages could not participate or complete the questionnaire.
12. The authors should be careful in a lot of details. For example, the numbering of Tables/Figures was not matched in results and in text. Line 185-187 lack reference for citation.
13. The discussion should place more emphasize on the similarity and differences on risk factors for LC and PACS. What these finding means and how should we proceed? Also, the authors did not explore the impact of vaccination and antiviral treatment on the studied associations, therefore cannot provide further information. It is good to bring this up. However, it will be beneficial to discuss how these factors could potentially affect the studied association, according to previous literature.
14. In limitation, the concepts on association and causation were mistakenly used. Causal effect cannot be directly estimated in observational study even ideally control for confounding factors.
15. "the time span between measurement and outcome is likely attenuating associations due to random misclassification" Why? Please clarify.

(Remarks on code availability)

Reviewer #3

(Remarks to the Author)

(Remarks on code availability)

Reviewer #4

(Remarks to the Author)

Summary:

The authors provide a descriptive analysis of the sociodemographic factors, biomarkers, and comorbidities that associate with long COVID (LC) and Post-Acute COVID-19 Syndrome (PACS) using a large sample from UK biobank data. The manuscript is well-written and clearly organized. Analysis is driven by a series of logistic regression models. Primary findings are that some biomarkers related to inflammation, cardiac function, and liver function are associated with LC and PACS.

Major comments:

1. The manuscript's overall contribution to LC/PACS research is unclear. While LC and PACS outcomes are shown to be associated with several demographic factors and biomarkers, it is not clear how these findings substantially advance the field or offer new insights. This study was undertaken as something of a 'fishing expedition' and the primary purpose is descriptive and hypothesis-generating. Most of the findings are entirely consistent with what is known and there is little interpretation in the discussion that generates new hypotheses to test.
2. The authors should comment more on the comparability of the case and control cohorts. After multivariable adjustment, many of the associations originally observed do not persist, and the remaining effect sizes are small. A notable portion of the manuscript is dedicated to crude associations that are later shown not to be significant after multivariable adjustment (see line 125). Similarly, several associations are significant for a lenient case definition of LC (only a single symptom required) only to lose significance when a stricter definition is used (as noted by the authors in lines 167-168). These limitations call into question the clinical relevance and utility of the results.
3. Because of multiple testing, it is advisable to apply a correction such as Benjamini-Hochberg to control false discovery rate (FDR). This might further reduce the number of significant findings that remain after adjustment.
4. The interpretation of results rests entirely on a series of regression models. There could be more information provided on the modeling approach. LASSO is a regularization technique but does not describe the model specification itself. Since the effect sizes are presented as ORs, it appears the models used are logistic regression models. However the plots in Figure 1 that assess linearity examine the relationship between probability of outcome and biomarker values. Logistic models are specified such that the log-odds (rather than probability) are expressed as a linear combination of predictors, so the assessment of linearity was done incorrectly. Beyond visual assessment, goodness-of-fit tests like Hosmer-Lemeshow can be helpful as model diagnostics to identify problems with specification.

Minor comments:

- Ensure correct figures are referenced throughout the manuscript (e.g. "Figure 4C" in line 132 appears to refer to Figure 3B)
- Line 65: It seems "lower" should actually be "higher"
- Line 62 and 65: Based on the numbers in Table 1, there appear to be (rounding) errors of the numbers reported in the text (e.g. 51 -> 52%, 43 -> 44%, 56 -> 57%)
- Line 198: Remove additional space after "with"
- Consider removing the paragraph between lines 204-209, as it does not add information relevant to this manuscript and the lack of vaccination data is sufficiently addressed in lines 241-242 (also line 209: "affect to" should be "affects")

(Remarks on code availability)

Reviewer #5

(Remarks to the Author)

(Remarks on code availability)

Version 1:

Reviewer comments:

Reviewer #1

(Remarks to the Author)

The authors have successfully addressed my comments.

(Remarks on code availability)

I have reviewed the code and find it to be well-structured, clearly annotated, and easy to follow. The functions used appear appropriate for the intended purpose.

While I cannot fully verify the reproducibility of the results without access to the underlying data, this appears to be a usable and valuable resource for the community.

Reviewer #2

(Remarks to the Author)

The authors have addressed most of my comments, by modifying existing content or providing additional explanations. There are however some remaining issues to be clarified.

First, the comment on providing additional summarized sentences on post-covid19 conditions in Introduction was not well addressed, leaving the rationale for exploring these outcomes to be less motivated.

Second, I am still puzzled by the reasons for selection on biomarkers, which is due to availability of data. Evidence shall be provided showing their links with outcomes of interest.

Third, the authors should be clear about the concepts on association and causation, and use appropriate causal inference language throughout the manuscript, for example, use "associated instead of cause or lead to", in observational study.

(Remarks on code availability)

Reviewer #3

(Remarks to the Author)

(Remarks on code availability)

Reviewer #4

(Remarks to the Author)

We thank the authors for taking the time to address the reviewer critiques. There are several remaining concerns.

Regarding Figure 1. The x-axis label is missing, and the figure caption does not sufficiently explain what is being shown or how the test was performed. If the test is based on the log-odds of the outcome model, it is unclear why probabilities are being plotted. Clarification is needed to interpret the figure meaningfully.

The addition of a citation does not adequately address the lack of clear definitions for LC and PACS in the main text. A plain-text definition of both cohorts should be included in the main body of the paper (not just in Supplementary Figure 1), as it is essential for readers to understand the cohort stratification underlying the analysis.

The order of figures remains confusing. Figure 1 does not present new results, and the study design does not appear until Figure 4. Reordering the figures to align more logically with the narrative would improve clarity.

(Remarks on code availability)

Reviewer #5

(Remarks to the Author)

(Remarks on code availability)

We thank the reviewers for their careful evaluation of our manuscript. Please find our responses to each comment attached. Kindly note that the line numbers refer to the clean version of the manuscript.

Reviewer #1 (Remarks to the Author):

This case-control study used UK Biobank data to examine clinical factors associated with Long COVID. A range of sociodemographic factors, biomarkers, and comorbidities were included. Strengths of the study included the incorporation of both self-reported and hospital-recorded symptoms, as well as the extensive biomarker data available in the Biobank data. The data are carefully curated, and the manuscript is well written. However, I have the following concerns.

1. The authors classified Post-COVID-19 Conditions (PCC) into two main categories: Long COVID and post-acute complications of SARS-CoV-2 infection (PACS) to distinguish between self-reported and more severe thromboembolic or cardiovascular complications. Specifically, Long COVID is defined as symptoms reported by patients themselves beyond 30 days, which differs from how Long-COVID is defined by the WHO and many other studies. While the operational definition of Long COVID do vary across studies, it would be helpful to briefly review how long-COVID has been previously defined and measured in the literature, given the growing body of research on this topic.

Thank you very much for this comment. We have now clarified in the main manuscript that our definition slightly differs from the one defined by the WHO, and briefly discuss the 30-day threshold (lines 307-312):

“The infected participants who had one or more WHO-listed LC symptoms beyond the acute period (30 day) were classified as cases, whereas others were included as controls. Notice that our definition differs slightly from that of the WHO, which defines LC as symptoms persisting beyond 90 days post-infection. At the time this study was conducted, there was no universally accepted definition. Previous studies have used a 30-day threshold, and current NHS guidance advice individuals to consult a GP if symptoms persist beyond 30 days³⁰”

Additionally, we have added a new sensitivity analysis, where we used the 90-day threshold (please, see sensitivity analysis 4). Direction of the associations remained similar, despite some differences in the statistical significance (Supplementary Figure 9).

2. Only two UK studies were discussed in the Introduction as prior evidence. It would be more informative to summarize existing evidence from a higher level by including findings from systematic review and meta-analysis (some of which the authors have already cited).

Thank you very much for highlighting this. We agree with the reviewer that higher level of evidence should be briefly summarised in the introduction. We have replaced the third paragraph with the following one, with the aim to summarise Tsampasian *et al.* systematic review and meta-analysis study (lines 31-36):

“Previous studies have aimed to characterise patients with LC⁷. Tsampasian et al. conducted a systematic review and meta-analysis of 41 studies to explore the risk factors of PCC in adult patients. Socio-demographics such as female sex, age, high BMI, and smoking were associated with an increased risk of PCC. The presence of comorbidities like anxiety or depression, asthma, chronic obstructive pulmonary disease (COPD), diabetes, immunosuppression, and ischaemic heart disease were also linked to an increased risk for PCC.”

3. PPC typically includes both the continuation of initial symptoms and the emergence of new symptoms following the initial infection. The authors included individuals with PACS diagnosis recorded between 30 days and 1-year post-infection, excluding those with pre-existing symptoms and those with diagnosis within the first 30 days. The analytical sample is therefore presents a highly selective group - those who did not experience any symptoms within 30 days after the infection. Factors associated with developing new symptoms after 30 days post-infection may be substantially different from those associated with persistence of symptoms. The rational for excluding people with persistent symptoms should be clarified.

Thank you for highlighting this point. We agree with the reviewer that our selection criteria define a specific subset of participants. However, this approach was necessary due to data limitations. In response to the issue of excluding those that reported symptoms within 30 days after infection, as we only had information of the symptoms at a single time point (i.e., when participants completed the questionnaire), it was not possible to determine whether those that answered the questionnaire within 30 days after infection, they would develop Long Covid afterwards. As a result, we had to exclude the participants who completed the questionnaire within 30 days of infection, as we could not reliably classify them as either cases or controls.

Regarding the issue of pre-existing symptoms, we aimed to exclude participants who reported symptoms prior to their COVID-19 infection, as these symptoms were assumed to be unrelated to the infection and likely attributable to other causes. However, we did not exclude participants who reported symptoms within the first thirty days after infection. We have now rephrased Supplementary Note 2 to clarify this point and have also included a graph to visually support our approach.

“The Health and well-being online questionnaire assessed the length of each symptom by asking people for how long they have been suffering the specific condition. The answer could be either (1) less than two weeks, (2) two to four weeks, (3) four to twelve weeks, (4) more than twelve weeks, (5) do not know, or (6) prefer not to answer. If participants reported options (1), (2), (5), (6), we assumed that the patient did not have pre-existing symptoms. If participants reported option (3) we assumed that the symptoms started four weeks before answering the questionnaire (28 days). If participants reported option (4), we directly assumed that the participants had pre-existing symptoms, as the date when the symptoms started could not be determined. Afterwards, we excluded those participants who reported option (4) or those whose date of answering the questionnaire minus the number of days they have been suffering the symptoms was before the date of the infection.”

4. The definition of Long COVID relies heavily on self-reported symptoms, with durations spanning from less than 2 weeks to over 12 weeks, which introduces potential recall bias. When a relative coarse measure of symptom duration is used in combination of a specific date of infection, the risk of misclassification is high and need to be discussed.

Thank you for pointing to this. Indeed, our study can suffer from recall bias. We have now discussed this in the limitations section (lines 252-255):

“Fifth, patient-reported-outcomes (PROs) are inherently subjective, which may result in misclassification of LC cases due to variability in symptom reporting. Additionally, the self-reported nature of the questionnaire can introduce recall bias, as participants may not accurately remember the symptoms or their length.”

5. Self-reported measures may also be associated with psychological factors and the presence of neuropsychiatric and mood disorders. Only dementia was included among comorbidities in the analysis. It would be helpful for the authors to explain how comorbidities were selected.

Thank you for highlighting this. We agree with the reviewers that this should be explained with detail in the manuscript. We have now added more information in the Methods section/Covariates and outcomes/Candidate covariates (lines 328-329):

“Comorbidities were selected based on Charlson Comorbidity Index⁴⁰ and phenotyped using linked HES data.”

The cited paper is the following one: A new method of classifying prognostic comorbidity in longitudinal studies: Development and validation - ScienceDirect and develops how the Charlson Comorbidity Index is calculated.

6. A series of exploration data analysis has been performed, including linearity assessments as shown in Figure 1. However, in logistic regression, the assumption of linearity applies to the relationship between the explanatory variables and the logit (log-odds) of the outcome – not the probability itself. Since the y-axis in Figure 1 depicts the probability of having the outcome, the figure may be misleading. It is also not clear how this test is performed. The authors should further clarify their approach and verify the validity of the test.

Thank you very much for this comment. We acknowledge that our approach is currently not clearly explained. We have added a note in Figure 1 legend clarifying it:

“Figure 1. Linearity assessment. (A) Evaluation of the non-linear relationship between the biomarkers and Long COVID using a natural cubic spline model. (B) Evaluation of the non-linear relationship between the biomarkers and PACS using a natural cubic spline model. **Note: Notice that the non-linear test was performed between the log-odds of the outcome model, not the probability of the outcome itself.”**

And we clarified our approach in the methods section (lines 342-347):

“We explored potential non-linearities between each biomarker and both study outcomes using natural cubic spline curves (adjusted using 3 knots). We compared the natural cubic spline model to the logistic regression model, and used ANOVA to assess whether there was a statistically significant difference between the two models. A p-value < 0.05 was used to indicate that the non-linear model

provided a better fit to the data, suggesting a potential non-linear association between the biomarker and the logit (log-odd) function of the study outcome.”

And in the results section (lines 70-74):

“We explored potential non-linear relationships between each biomarker and the study outcomes using natural cubic spline curves. For C-reactive protein and HDL cholesterol, the non-linear cubic spline model provided a better fit for the logit function of LC compared to a linear model (see Figure 1A). For the other biomarkers, the linear logistic regression model was a better fit.”

7. Variable selection was performed stepwise by fitting three LASSO regressions. What is the rationale for not including all variables in a single regression as biomarkers may also be correlated with comorbidities? Is this because of a concern of power? The rationale for this approach should be explained more clearly.

Thank you very much for your comment. We conducted three logistic regression with LASSO penalisation, one for each group of covariates, to identify as many relevant risk factors as possible. In the crude analysis, where each factor is assessed independently, correlations between biomarkers and comorbidities should not affect the results. However, multicollinearity may influence the adjusted model, where all risk factors are modelled together. To address this, we calculated the variance inflation factor to assess collinearity risk and conducted sensitivity analysis 1, where we excluded biomarkers known to be highly correlated with specific comorbidities. We have tried to make this more clear in the main manuscript. Please see lines 349-351:

“We used logistic regression with LASSO penalisation to select key variables for modelling. Three separate regressions were performed, one for each group of covariates, with the aim to identify as many risk factors as possible: sociodemographic, biomarkers, and comorbidities.”

Reviewer #1 (Remarks on code availability):

The code is clearly annotated. There is a README file, but it requires login to see the details. I am not able to run the code as I don't have access to the data, hence I cannot assess the reproducibility of the code.

Thank you for your comment. GitHub webpage requires to login to see the details, which is out of our domain. However, if one has an account, the code is publicly available.

Reviewer #2 (Remarks to the Author):

This is a case-control study utilizing COVID-19 infected individuals in the UK Biobank to explore the association of sociodemographic factors, clinical biomarkers, and comorbidities with the risk of Long COVID (LC) and Post-Acute COVID-19 Syndrome (PACS). The authors found an association between inflammation, cardiovascular, and liver function markers to be associated with risk of LC, and an addition of high levels of HbA1c, cystatin C, and urate to be association with PACS. This study benefits from the comprehensive statistical analyses and fruitful covariates to be included, while the summary of pre-existing evidence and result interpretation can be greatly improved. I do have several comments for the authors to consider:

1. It is suggested to provide a more comprehensive summary on post-covid19 conditions in the

Introduction. The authors mentioned that risk of PCC was potentially independent of the severity of the acute COVID-19 infection. I however found some evidence showing risk of PCC was higher among individuals recovering from the most severe form of acute COVID-19 infection (Magnúsdóttir I, 2022; Shen Q, 2023).

Thank you for highlighting this. We acknowledge that our phrasing was unclear. Our intention was to explain that the risk of developing post-COVID conditions may not depend solely on the severity of the acute infection, but rather on a wide range of contributing factors. We have now revised the sentence accordingly (lines 26-27):

“with risk for PCC potentially not solely dependent on the severity of the acute COVID-19 infection.”

2. It appears unclear to me how LC and PACS, the two primary outcomes, was defined by definitions. It seems there're overlapped symptoms in these two outcomes? I do not find evidence showing a clear definition or clear thresholds to separate these outcomes by checking the cited references. I do not follow why and how PACS refers to more severe thromboembolic or cardiovascular complications. Please provide justifications on why studying these two outcomes in this study is important, and what additional evidence could be provided in current study for LC in addition to existing studies.

Thank you for your comment. We agree that we need to clarify the distinction between LC and PACS. While these outcome may share overlapping symptoms, we considered it important to differentiate them, as they likely reflect distinct underlying mechanisms. LC typically refers to persistent symptoms without a clear alternative diagnosis, whereas PACS may encompass more severe post-acute sequelae.

To clarify this, we have now cited Peluso *et al.*, Mechanisms of long COVID and the path toward therapeutics: Cell in the introduction (line 28, ref Num 4). In this study, the authors use PACS as an umbrella term for all post-acute sequelae of SARS-CoV-2 infection, while Long Covid is used to describe symptoms that cannot be explained by an alternative diagnosis (see Figure 1 in the referenced paper).

3. I am not following the Suppl.Fig1 flowchart. What's the difference between “restrict to participants that answered the health and well-being questionnaire”, and “restrict to participants that answered yes/no to ...questionnaire”? If this is to exclude missing value or prefer not to answer, it will be helpful to clarify numbers and reasons for this exclusion. The authors may consider to justify why restrict to the latest record for covid-19 infection for LC cohort while consider the earliest infection for PACS cohort? Suppl. Figure 11 LC cohort can be followed, but PACS cohort is very hard to understand. Suggest to add description or notes under this picture to help understanding.

Thank you very much for your comment. By “restricting to participants who answered the health and well-being questionnaire”, we refer to the ~200,000 UK Biobank participants who took part in the Health and Well-Being study. The phrase “restrict to participants who answered yes/no to the questions... questionnaire”, refers to excluding those who selected “Do not know” or “Prefer not to answer”, as such responses could lead to misclassification. Based on this last criteria, 7,490 participants were excluded from 54,283. We have now added a clarifying note (in Supplementary Figure 1 legend) to clearly distinguish between these two exclusion criteria:

“Note 1. “Restrict to participants that answered the health and well-being web-questionnaire” refers to including only UK Biobank participants who completed the Health and Well-being study.

Note 2. “Restrict to participants that answered yes/no to all the questions from the health and well-being web-questionnaire” means excluding those who answered “Do not know” or “Prefer not to answer” for any symptoms-related question.”

Regarding the restriction to the latest COVID-19 infection record, this was done to ensure that the time elapsed since infection could be calculated at the moment participants completed the questionnaire. This allowed us to determine whether they met the ≥ 30 day threshold required to be classified as having Long COVID. In contrast, for the PACS cohort, no questionnaire data were used. Therefore, the earliest infection record was used solely to determine age at infection, without affecting case or control classification.

We have added a note in Supplementary Figure 11 legend to explain in detail how the PACS cohort was created.

4. I am confused by the definition of pre-existing symptoms. The authors mentioned “If participants reported options (1), (2), (4), (5), we assumed that the patient has not have persistent symptoms. If participants reported option (3) we assumed that the persistent symptoms had a length of four weeks (28 days). If participants reported option (4), we directly assumed that the participants have persistent symptoms.” Where exactly option (4) participants were grouped to?

Thank you very much for highlighting this. Index, we noticed that there is a mistake in the text and should be like this:

“If participants reported options (1), (2), (5), (6), we assumed that the patient did not have pre-existing symptoms. If participants reported option (3) we assumed that the symptoms started four weeks before answering the questionnaire (28 days).”

This has now been corrected in the supplementary file.

5. I am a little puzzled by the clinical implementation of this study. How can biomarkers and characteristics assessed over 10 years prior to COVID-19 infection being able to be used for prediction of Long COVID syndrome, or help identify high risk groups?

Thank you for your comment. We acknowledge that the long-time span between baseline measurements and COVID-19 infection may attenuate associations due to potential misclassification, likely biasing results to the null through random error. However, our findings are consistent with existing literature. Moreover, the biomarker patterns observed align with those from more recent comorbidity data, providing biologically plausible explanations for how pre-existing health conditions may contribute to Long COVID. That being said, we agree that further studies, specifically designed to assess causal relationships, are necessary.

6. The authors lack justifications on selection of risk factors or biomarkers. Why include these covariates and the supporting evidence?

Thank you for your comment. Biomarkers were selected based on their availability in UK Biobank. Comorbidities were selected based on the Charlson Comorbidity Index. We have now added a sentence in the Methods section/Covariates and outcomes/Candidate covariates (lines 328-329):

“Comorbidities were selected based on Charlson Comorbidity Index⁴⁰ and phenotyped using linked HES data.”

The cited paper is the following one: A new method of classifying prognostic comorbidity in longitudinal studies: Development and validation - ScienceDirect and develops how the Charlson Comorbidity Index is calculated.

7. The CDC has recommended a BMI less than 18.5 to be underweight and 18.5 to 25 to be “healthy weight”. Any reason not to consider underweight in the categorization of BMI?

Thank you for your comment. We did not include the underweight category in our main analyses due to insufficient sample size. In the Long COVID cohort (8,668 individuals), only 38 were classified as underweight according to CDC criteria. Similarly, in the PACS cohort (108,407 individuals), only 466 individuals fell into this category. Given these small numbers, any estimates would be highly unstable and potentially misleading. We have added this clarification to Supplementary Note 4:

“Reference group is average weight. Underweight group was not considered due to insufficient sample size.”

8. Variables with more than 50% of missing data were excluded from further analysis. How missing value was handled for those variables with less than 50% missing data?

Thank you for your question. We used multiple imputation for those variables with less than 50% of missing data. This is well explained in Supplementary Note 3, but we agree with the reviewer that some context needs to be added in the main manuscript. We have now added a sentence in Methods/Statistical analysis/Exploratory data analysis (lines 335-338):

“Variables with more than 50% of missing data were excluded from further analysis. Multiple imputation was used for variables with less than 50% of missing data. For continuous variables, we calculated the mean, standard deviation, quantiles, and their distribution. For categorical variables, we report proportions and depict the data using histograms.”

9. The authors applied LASSO regression for modelling. It will be better to provide more details how variables were fitted in the model, and provide appropriate citations. For instance, why select one for each group of covariates? what parameters were used for spline function?

Thank you very much for this comment. We agree with the reviewer that more details should be added regarding these regressions. We have now specified that we used four degrees of freedom for the natural cubic spline curves (line 342-343):

“We explored potential non-linearities between each biomarker and both study outcomes using natural cubic spline curves (adjusted using 3 knots).”

We also specified that we used a logistic regression with LASSO penalisation for the variable selection (line 348):

“We used logistic regression with LASSO penalisation to select key variables for modelling.”

10. Why fit logistic regression models separately for the three categories of covariates? In sensitivity analyses, why exclude several biomarkers to be associated with some comorbidities, any reference or evidence? Using three symptoms in sensitivity analyses for LC definition should be justified.

Thank you very much for your comment. We have answered a similar question from reviewer 1. We conducted three logistic regression with LASSO penalisation, one for each group of covariates, to identify as many relevant risk factors as possible. In the crude analysis, where each factor is assessed independently, correlations between biomarkers and comorbidities should not affect the results. However, multicollinearity may influence the adjusted model, where all risk factors are modelled together. To address this, we calculated the variance inflation factor to assess collinearity risk and conducted sensitivity analysis 1, where we excluded biomarkers known to be highly correlated with specific comorbidities. We have tried to make this more clear in the main manuscript. Please see lines 349-351:

“We used logistic regression with LASSO penalisation to select key variables for modelling. Three separate LASSO regressions were performed, one for each group of covariates, with the aim to identify as many risk factors as possible: sociodemographic, biomarkers, and comorbidities.”

We used three symptoms in the sensitivity analysis for LC definition to minimise the risk of misclassification. As LC was determined by self-reported symptoms. We have now added a sentence in the Methods section (lines 374-377):

“Thirdly, we repeated the main analysis using an alternative LC definition where participants were classified as cases if they reported at least three WHO proposed symptoms. Given the reliance on self-reported data, this approach aimed to reduce the risk of misclassification. Those with fewer than two symptoms or who did not report any symptoms were classified as controls.”

11. In discussion, the authors mentioned an opposite trend of age to be associated with lower risk of LC, and explained that differential mortality rates could be the reason behind. The authors may also consider the reporting bias in questionnaire, where the most severe patients of older ages could not participant or complete the questionnaire.

Thank you very much for highlighting this. We have added this point to the discussion (lines 189-194):

“One possible explanation for this discrepancy could be either differential mortality rates in older groups compared to younger ones, and differences in defining LC phenotypes; most prior studies relied on clinical diagnosis from primary or secondary care data, whereas our study used patient self-reported outcomes. Reporting bias may also play a role, as older individuals with more severe outcomes may have been less likely to complete the health and well-being questionnaire.”

12. The authors should be careful in a lot of details. For example, the numbering of Tables/Figures was not matched in results and in text. Line 185-187 lack reference for citation.

Apologies for this. We have now reviewed the manuscript and made sure that Tables and Figures matched with the corresponding numbering. We have also added the missing citation (please see line 187, ref. 7).

13. The discussion should place more emphasize on the similarity and differences on risk factors for LC and PACS. What these finding means and how should we proceed? Also, the authors did not explore the impact of vaccination and antiviral treatment on the studied associations, therefore cannot provide further information. It is good to bring this up. However, it will be beneficial to

discuss how these factors could potentially affect the studied association, according to previous literature.

Thank you very much for this comment. We have highlighted and discussed further on most prominent differences of risk factors: age (younger for LC, older for PACS) and sex (female for LC, male for PACS), specifically throughout the discussion section. Please see the added contents here and in the revised manuscript:

Lines 185-194: Discussion about age differences

“Notably, although old age has been an established risk factor for severe COVID-19 in the acute phase, its role in long-term outcomes remains unclear. A systematic review and meta-analysis of nine studies found that adults older than 40 had a higher risk of LC compared to younger adults⁷ However, our study found aging was only positively associated with the PACS phenotype. In contrast it was negatively associated LC. This is in line with our previous research¹¹ and other studies¹². One possible explanation for this discrepancy could be either differential mortality rates in older groups compared to younger ones, and differences in defining LC phenotypes; most prior studies relied on clinical diagnosis from primary or secondary care data, whereas our study used patient self-reported outcomes. Reporting bias may also play a role, as older individuals with more severe outcomes may have been less likely to complete the health and well-being questionnaire”.

Lines 194-204: Discussion about sex differences

“Importantly, we found females were more prone to LC and males to PACS. Similar results have been seen in previous studies, where the likelihood of developing long COVID syndrome was significantly higher in females compared to males¹³, although endocrine and renal disorders were higher among males. Generally, females mount stronger innate and adaptive immune responses to infections and vaccinations compared to males^{14,15}. This can be beneficial in clearing acute infections, but it may also predispose females to prolonged immune dysregulation and a higher risk of developing persistent inflammation or even autoimmune conditions. Instead, males are more likely to experience severe acute COVID-19, with higher rates of hospitalization, ICU admission, and death compared to females. This increased severity during the acute phase can lead to more significant organ damage, which predispose more males with PACS.

There are two paragraphs that discuss the impact of vaccination and antiviral treatment on the studied associations (lines 218-243):

“Our study focused on identifying determinants of LC and PACS, but it is important to consider the broader context of interventions that may influence these outcomes. For example, vaccination before infection has also been reported to partially mediate the risk of sequelae at 6 months by 15%²⁰. A recent study that conducted a multinational staggered cohort study²¹ suggested that vaccination could mitigate the risk of acute and subacute post COVID-19 venous thromboembolism and heart failure by a 40% and 30%, respectively, for 90-180 days post COVID-19 infection. However, we did not have data regarding vaccination so we could not explore how it affects the outcomes.

Additionally, the use of antivirals (molnupiravir or nirmatrelvir) within 5 days after infection has also been associated with lower risk of sequelae by 14% and 26%, respectively²². However, only patients at risk of progression to severe COVID-19 are eligible for these drug interventions during the acute phase, and their risk-benefit in a wider population with milder infection, as well as post-acute stage of disease, remain unclear. Importantly, while most post-COVID symptoms are more common following severe cases, the highest overall burden is still among those with a history of mild-to-

moderate COVID-19 illness, as they represent most infected individuals now and likely in the future. Our findings on potential biomarkers in general support the promise of drug development targeting inflammation, the immune system, or blood clots-pathways currently under clinical investigation²³. However, the efficacy of these approaches remains unpredictable due to the interplay and complexity of the underlying mechanisms^{4,24,25}.”

14. In limitation, the concepts on association and causation were mistakenly used. Causal effect cannot be directly estimated in observational study even ideally control for confounding factors.

Thank you for your comment. We have now removed the sentence “ideally controlling for confounding” and now it reads like this (lines 244-246):

“First, our analysis describes associations, not causal effects. Further research is needed to establish the nature of these associations.”

15. “the time span between measurement and outcome is likely attenuating associations due to random misclassification” Why? Please clarify.

Thank you for your comment. The time span between measurement of the biomarkers and COVID-19 infection may attenuate association due to potential misclassification, biasing results to the null through random error.

Reviewer #3 (Remarks to the Author):

Thank you very much for all your comments.

Reviewer #4 (Remarks to the Author):

Summary:

The authors provide a descriptive analysis of the sociodemographic factors, biomarkers, and comorbidities that associate with long COVID (LC) and Post-Acute COVID-19 Syndrome (PACS) using a large sample from UK Biobank data. The manuscript is well-written and clearly organized. Analysis is driven by a series of logistic regression models. Primary findings are that some biomarkers related to inflammation, cardiac function, and liver function are associated with LC and PACS.

Major comments:

1. The manuscript’s overall contribution to LC/PACS research is unclear. While LC and PACS outcomes are shown to be associated with several demographic factors and biomarkers, it is not clear how these findings substantially advance the field or offer new insights. This study was undertaken as something of a ‘fishing expedition’ and the primary purpose is descriptive and hypothesis-generating. Most of the findings are entirely consistent with what is known and there is little interpretation in the discussion that generates new hypotheses to test.

Thank you very much for this comment. The key strength of our study is the separate analysis of LC and PACS. While most of the previous studies have focused on evaluating these outcomes together,

our findings highlight that the associated risk factors may differ. This distinction contributes to a clearer understanding and definition of post-acute COVID-19 conditions.

This is highlighted in the strengths section of the discussion (lines 241-243):

“Second, we separately examined LC and PACS, being, to our knowledge, the first study to date to explore the key variables associated with these two outcomes independently. Importantly, we identified key differences in the associations between the variables and these outcomes.”

2. The authors should comment more on the comparability of the case and control cohorts. After multivariable adjustment, many of the associations originally observed do not persist, and the remaining effect sizes are small. A notable portion of the manuscript is dedicated to crude associations that are later shown not to be significant after multivariable adjustment (see line 125). Similarly, several associations are significant for a lenient case definition of LC (only a single symptom required) only to lose significance when a stricter definition is used (as noted by the authors in lines 167-168). These limitations call into question the clinical relevance and utility of the results.

Thank you very much for this comment. When referring to sociodemographic characteristics and comorbidities, we focused on adjusted results to account for potential confounding factors and better isolate the effect of each variable. However, we discussed crude results more extensively for biomarkers because we considered that they could be more informative, given the biological plausibility of many biomarkers as independent predictors. Additionally, although we tried to control for multicollinearity, residual correlation between biomarkers and comorbidities may result in less stable results.

Under the more stringent LC definition (requiring at least three symptoms), most of the association were no longer statistically significant, though their direction remained constant. This may be due to limited statistical power, given the significant imbalance between cases (3%) and controls (97%).

3. Because of multiple testing, it is advisable to apply a correction such as Benjamini-Hochberg to control false discovery rate (FDR). This might further reduce the number of significant findings that remain after adjustment.

Thank you very much for your comment. The Benjamini-Hochberg (BH) Procedure is indeed a widely used method to control the false discovery rate (FDR), and it is suitable for exploratory analyses where one is willing to accept some false positives in order to maintain the statistical power.

However, in our sensitivity analysis (num. 5), we preferred to use Bonferroni correction, which is known to be more conservative as BH. We decided to choose this method because the main goal of our sensitivity analysis was to assess the robustness of our findings when correcting by multiple testing. By applying Bonferroni correction, we ensured that any associations that remained statistically significant were highly unlikely to be false positives.

4. The interpretation of results rests entirely on a series of regression models. There could be more information provided on the modelling approach. LASSO is a regularization technique but does not describe the model specification itself. Since the effect sizes are presented as ORs, it appears the models used are logistic regression models. However the plots in Figure 1 that assess linearity examine the relationship between probability of outcome and biomarker values. Logistic models are specified such that the log-odds (rather than probability) are expressed as a linear combination of predictors, so the assessment of linearity was done incorrectly. Beyond visual assessment, goodness-of-fit tests like Hosmer-Lemeshow can be helpful as model diagnostics to identify problems with specification.

Thank you very much for this comment. We have now specified that we used a logistic regression with LASSO penalisation (line 349):

“We used logistic regression with LASSO penalisation to select key variables for modelling.”

Regarding the linearity assessment, we apologise as we acknowledge that we did not explain clearly our approach in the previous version of the manuscript. We have now clarified our approach in the methods sections:

“We explored potential non-linearities between each biomarker and both study outcomes using natural cubic spline curves (adjusted using 3 knots). We compared the natural cubic spline model to the logistic regression model, and used ANOVA to assess whether there was a statistically significant difference between the two models. A p -value < 0.05 was used to indicate that the non-linear model provided a better fit to the data, suggesting a potential non-linear association between the biomarker and the logit (log-odd) function of the study outcome.”

So, essentially, we compared the logit function of the outcome using either cubic splines or logistic regression. As Figure 1 can be misleading (as reviewer 1 also pointed out), we added a note clarifying it:

*“Figure 1. Linearity assessment. (A) Exploration of the non-linear relationship between the biomarkers and Long COVID. (B) Exploration of the non-linear relationship between the biomarkers and PACS. **Note:** Notice that the non-linear test was performed between the log-odds of the outcome model, not the probability of the outcome itself.”*

Minor comments:

- Ensure correct figures are referenced throughout the manuscript (e.g. “Figure 4C” in line 132 appears to refer to Figure 3B)

Thank you for highlighting this. We have now corrected this.

- Line 65: It seems “lower” should actually be “higher”

Thank you for highlighting this. We have corrected this.

- Line 62 and 65: Based on the numbers in Table 1, there appear to be (rounding) errors of the numbers reported in the text (e.g. 51 -> 52%, 43 -> 44%, 56 -> 57%)

Thank you for highlighting this. This has now been corrected.

- Line 198: Remove additional space after “with”

Thank you for highlighting this. The additional space has been removed.

- Consider removing the paragraph between lines 204-209, as it does not add information relevant to this manuscript and the lack of vaccination data is sufficiently addressed in lines 241-242 (also line 209: “affect to” should be “affects”)

Thank you very much for your comment. We decided to keep the paragraph as another reviewer

highlighted that the implication of vaccination and antiviral treatments should be discussed. We have corrected “affect to” to “affects”.

Reviewer #5 (Remarks to the Author):

Thank you for all your comments.

We thank the reviewers for their careful evaluation of our manuscript. Please find our responses to each comment attached. Kindly note that the line numbers refer to the clean version of the manuscript.

Reviewer #1 (Remarks to the Author):

I have reviewed the code and find it to be well-structured, clearly annotated, and easy to follow. The functions used appear appropriate for the intended purpose.

While I cannot fully verify the reproducibility of the results without access to the underlying data, this appears to be a usable and valuable resource for the community.

We thank the reviewer for their positive assessment of our code.

Reviewer #2 (Remarks to the Author):

The authors have addressed most of my comments, by modifying existing content or providing additional explanations. There are however some remaining issues to be clarified.

First, the comment on providing additional summarized sentences on post-covid19 conditions in Introduction was not well addressed, leaving the rationale for exploring these outcomes to be less motivated.

We thank the reviewer for this important feedback and apologise for the previous omission. We have now expanded the second paragraph of the introduction, which may provide a stronger rationale for exploring these distinct outcomes.

“PCC can be classified into two main categories⁴⁻⁶: Long COVID (LC), and post-acute complications of SARS-CoV-2 infection (PACS). LC is defined by the persistence or onset of COVID-19 related symptoms beyond one to three months after the initial infection. Common symptoms include fatigue, shortness of breath, and other symptoms that can significantly impact day-to-day functioning. In contrast, PACS typically refers to more severe complications emerging in the same time frame, such as thromboembolic or cardiovascular events including angina, myocardial infarction, or pulmonary embolism.”

Second, I am still puzzled by the reasons for selection on biomarkers, which is due to availability of data. Evidence shall be provided showing their links with outcomes of interest.

Thank you very much for this comment. Our study was initially designed as a hypothesis-free, data-driven analysis. The primary goal was to explore a wide range of potential risk factors without being constrained by pre-existing hypotheses. This could enable us to identify novel associations for the complex, new conditions like LC and PACS.

We have now emphasised this at the end of the introduction section:

*“We therefore leveraged linked UK Biobank data to perform a **hypothesis-free analysis** and explore the associations and interplay between sociodemographic factors, biomarkers, and comorbidities and the risk of LC and PACS.”*

In the Methods/Covariates and outcomes/Covariates section:

“Several socio-demographics characteristics (6), clinical biomarkers (30) and comorbidities (19) were pre-specified for the analysis of associations with LC and PACS and modelled

separately. Socio-demographic characteristics and biomarkers were extracted from the baseline UK Biobank assessment (conducted between 2006 and 2010), where information was collected through touchscreen questionnaires and biological samples. Biomarkers were selected for the analysis based on their availability in UK Biobank. The UK Biobank selected these biomarkers because they represent established risk factors for disease, and are clinical diagnostic measures. Comorbidities were selected based on Charlson Comorbidity Index⁴⁰ and phenotyped using linked HES data.”

And at the beginning of the discussion section:

“We conducted a comprehensive data-driven hypothesis-free analysis to explore the association of 55 prespecified candidate determinants, including 6 socio-demographics, 30 clinical biomarkers, and 19 comorbidities, with the risk of LC and PACS.”

Third, the authors should be clear about the concepts on association and causation, and use appropriate causal inference language throughout the manuscript, for example, use "associated instead of cause or lead to", in observational study.

We apologise for the lack of clarity about the concept “association”. We have tried to be very careful when using the terms “cause” or “lead to”. That is why, when reporting our results we have used the terms “associated with” and “linked to”. Throughout the discussion, when referencing other observational studies, we have also used these terms.

Reviewer #3 (Remarks to the Author):

Thank you very much for reviewing the manuscript.

Reviewer #4 (Remarks to the Author):

We thank the authors for taking the time to address the reviewer critiques. There are several remaining concerns.

Regarding Figure 1. The x-axis label is missing, and the figure caption does not sufficiently explain what is being shown or how the test was performed. If the test is based on the log-odds of the outcome model, it is unclear why probabilities are being plotted. Clarification is needed to interpret the figure meaningfully.

We have made the following improvements:

Axis Label: The missing x-axis label has been added and is now correctly labelled “Z-score of the biomarker” in the revised Figure 2.

Probabilities vs. Log-Odds: The reviewer is correct that the non-linearity test is performed on the log-odds scale. This is our deliberate choice to improve interpretability for a broader audience, as probabilities (on a 0-1 scale) could be more intuitive. We have attached the figures based on the log-odds of the outcomes, which shows that the actual shapes are similar to the ones provided for the probability of the outcome.

We have further added clarifications in the figure legend, to avoid confusion:

“Figure 1. Linearity assessment. (A) Evaluation of the non-linear relationship between the biomarkers and Long COVID using a natural cubic spline model. (B) Evaluation of the non-linear relationship between the biomarkers and PACS using a natural cubic spline model.

Note: Notice that the non-linear test was performed between the log-odds of the outcome model, not the probability of the outcome itself. However, the probability of the outcome is plotted as it is more intuitive and accessible for interpretation. Notice that the shape of the curve is similar when plotted on the log-odds scale”.

The addition of a citation does not adequately address the lack of clear definitions for LC and PACS in the main text. A plain-text definition of both cohorts should be included in the main body of the paper (not just in Supplementary Figure 1), as it is essential for readers to understand the cohort stratification underlying the analysis.

Thank you very much for highlighting this. We apologise for not addressing the comment. Similarly to what Reviewer #1 suggested, we have now expanded the second paragraph of the introduction including a brief summary of Long COVID and PACS:

“PCC can be classified into two main categories⁴⁻⁶: Long COVID (LC), and post-acute complications of SARS-CoV-2 infection (PACS). LC is defined by the persistence or onset of COVID-19 related symptoms beyond one to three months after the initial infection. Common symptoms include fatigue, shortness of breath, and other symptoms that can significantly impact day-to-day functioning. In contrast, PACS typically refers to more severe complications emerging in the same time frame, such as thromboembolic or cardiovascular events including angina, myocardial infarction, and pulmonary embolism.”

The order of figures remains confusing. Figure 1 does not present new results, and the study design does not appear until Figure 4. Reordering the figures to align more logically with the narrative would improve clarity.

Thank you for pointing this out. We have moved figure 4 to the results section, and now is figure 1.

Reviewer #5 (Remarks to the Author):

Thank you very much for reviewing the manuscript.